# Mixture of Experts Meets Prompt-Based Continual Learning

**Minh Le**[3]    **An Nguyen**[2*]    **Huy Nguyen**[1*]    **Trang Nguyen**[3*]

**Trang Pham**[3*]    **Linh Van Ngo**[2]    **Nhat Ho**[1]

[1] The University of Texas at Austin
[2] Hanoi University of Science and Technology
[3] VinAI Research

## Abstract

Exploiting the power of pre-trained models, prompt-based approaches stand out compared to other continual learning solutions in effectively preventing catastrophic forgetting, even with very few learnable parameters and without the need for a memory buffer. While existing prompt-based continual learning methods excel in leveraging prompts for state-of-the-art performance, they often lack a theoretical explanation for the effectiveness of prompting. This paper conducts a theoretical analysis to unravel how prompts bestow such advantages in continual learning, thus offering a new perspective on prompt design. We first show that the attention block of pre-trained models like Vision Transformers inherently encodes a special mixture of experts architecture, characterized by linear experts and quadratic gating score functions. This realization drives us to provide a novel view on prefix tuning, reframing it as the addition of new task-specific experts, thereby inspiring the design of a novel gating mechanism termed Non-linear Residual Gates (NoRGa). Through the incorporation of non-linear activation and residual connection, NoRGa enhances continual learning performance while preserving parameter efficiency. The effectiveness of NoRGa is substantiated both theoretically and empirically across diverse benchmarks and pretraining paradigms. Our code is publicly available at `https://github.com/Minhchuyentoancbn/MoE_PromptCL`.

## 1 Introduction

Humans possess a remarkable ability to learn continuously by integrating new skills and knowledge while retaining past experiences. However, current AI models often fail to retain this ability. Unlike humans, they often suffer from *catastrophic forgetting* [28, 30, 32, 38], a phenomenon where they struggle to retain knowledge from previous tasks while learning new ones. Inspired by human learning, Continual Learning [2, 28, 43, 1, 12] is an ongoing field that aims to train a model across a sequence of tasks while mitigating this challenge. Traditional continual learning methods often rely on storing past data for fine-tuning, which can raise concerns about memory usage and privacy [5, 39, 51]. To address these limitations, prompt-based approaches have emerged as a promising alternative within rehearsal-free continual learning. By attaching prompts - small sets of learnable parameters - to a frozen pre-trained model, these approaches enable efficient adaptation to new tasks with minimal modifications to the underlying model [56, 26, 61]. The effectiveness of prompt-based methods has been demonstrated by several recent works achieving state-of-the-art performance on various continual learning benchmarks [49, 53, 54].

While prompt-based methods have demonstrably achieved impressive results, their emphasis largely lies on prompt utility, leaving a gap in our theoretical comprehension of their effectiveness. This

---

*Equal contribution.

38th Conference on Neural Information Processing Systems (NeurIPS 2024).

absence of a theoretical foundation hinders our ability to further refine and optimize these methods. In this work, we offer a new perspective by focusing on prefix tuning [26] and its connection to mixture of experts models [19, 15, 13, 11]. We demonstrate that self-attention blocks in Vision Transformers [8] implicitly encode a special mixture of experts architecture, revealing a surprising connection between these seemingly disparate concepts. Leveraging this connection, we propose that applying prefix tuning within pre-trained models can be interpreted as introducing new experts. The newly introduced experts collaborate with the pre-trained experts, facilitating efficient adaptation of the model to new tasks.

Drawing insights from this analysis, we observe that the original prefix tuning suffers from suboptimal sample efficiency, requiring a substantial amount of data for reasonable parameter estimation. To address this challenge, we propose a novel gating mechanism termed Non-linear Residual Gates (NoRGa). This architecture integrates non-linear activation functions and residual connections within the gating score functions. Our work focuses on improving within-task prediction accuracy, a key component of continual learning performance as identified in previous research [22, 49]. We posit that NoRGa can enhance this aspect, which contributes to improved overall continual learning performance while maintaining parameter efficiency. We further provide theoretical justification for this improvement, demonstrating how NoRGa accelerates parameter estimation rates.

**Our contributions** can be summarized as follows: (1) We reveal a novel connection between self-attention and a mixture of experts, providing a fresh perspective on prompt-based continual learning approaches; (2) Leveraging this insight, we propose *Non-linear Residual Gates (NoRGa)*, an innovative gating mechanism that enhances continual learning performance while maintaining parameter efficiency, and provide a theoretical justification for this improvement; (3) Extensive experiments across various continual learning benchmarks and pre-training settings demonstrate that our approach achieves state-of-the-art performance compared to existing methods.

**Notation.** For any $n \in \mathbb{N}$, we denote $[n]$ as the set $\{1, 2, \ldots, n\}$. Next, for any set $S$, we let $|S|$ stand for its cardinality. For any vector $u := (u_1, u_2, \ldots, u_d) \in \mathbb{R}^d$ and $\alpha := (\alpha_1, \alpha_2, \ldots, \alpha_d) \in \mathbb{N}^d$, we let $u^\alpha = u_1^{\alpha_1} u_2^{\alpha_2} \ldots u_d^{\alpha_d}$, $|u| := u_1 + u_2 + \ldots + u_d$ and $\alpha! := \alpha_1! \alpha_2! \ldots \alpha_d!$, while $\|u\|$ stands for its 2-norm value. Lastly, for any two positive sequences $\{a_n\}_{n \geq 1}$ and $\{b_n\}_{n \geq 1}$, we write $a_n = \mathcal{O}(b_n)$ or $a_n \lesssim b_n$ if $a_n \leq C b_n$ for all $n \in \mathbb{N}$, where $C > 0$ is some universal constant. The notation $a_n = \mathcal{O}_P(b_n)$ indicates that $a_n / b_n$ is stochastically bounded.

## 2 Background and Related Works

We first provide background and related works on continual learning. Then, we define the attention mechanism, followed by discussions on prompt-based continual learning and mixture of experts.

**Continual Learning (CL)** addresses the challenge of training a model incrementally on a sequence of $T$ tasks, denoted by $\mathcal{D} = \{\mathcal{D}_1, ..., \mathcal{D}_T\}$. Each task's training data $\mathcal{D}_t = \{(\boldsymbol{x}_i^{(t)}, y_i^{(t)})\}_{i=1}^{N_t}$ contains pairs of input sample $\boldsymbol{x}_i^{(t)} \in \mathcal{X}^{(t)}$, and corresponding label $y_i^{(t)} \in \mathcal{Y}^{(t)}$. Notably, the class labels are distinct for each task, *i.e.,* $\mathcal{Y}^{(t)} \bigcap \mathcal{Y}^{(t')} = \varnothing, \forall t \neq t'$. Consider a neural network with a backbone function $f_\theta$ and an output layer $h_\psi$. The model predicts a label $\hat{y} = h_\psi(f_\theta(\boldsymbol{x})) \in \mathcal{Y} = \bigcup_{t=1}^T \mathcal{Y}^{(t)}$, where $\boldsymbol{x} \in \mathcal{X} = \bigcup_{t=1}^T \mathcal{X}^{(t)}$ is an unseen test sample from arbitrary tasks. Importantly, during training on a new task, the model can only access the current data, without access to data from previous tasks. Prior approaches often rely on storing past task samples for training on new tasks, raising concerns regarding storage and privacy [5, 6, 39, 51, 59].

Our work focuses on the class-incremental learning (CIL) setting, where task identities are not provided during inference, unlike in task-incremental learning (TIL) [46]. A recent theory by [22] analyzes the CIL objective by decomposing the probability of a test sample $\boldsymbol{x}$ of the $j$-th class in task $t$ into two probabilities:

$$P(\boldsymbol{x} \in \mathcal{X}_j^{(t)} | \mathcal{D}) = P(\boldsymbol{x} \in \mathcal{X}_j^{(t)} | \boldsymbol{x} \in \mathcal{X}^{(t)}, \mathcal{D}) P(\boldsymbol{x} \in \mathcal{X}^{(t)} | \mathcal{D}), \qquad (1)$$

where the first term involves within-task prediction (WTP) and the second term pertains to task-identity inference (TII). This equation highlights that by improving either the WTP performance or the TII, we can consequently improve the overall CIL performance, as shown in [22, 49].

**Attention Mechanism.** Within the Transformer architecture, the attention mechanism plays a crucial role. One prevalent variant is scaled dot-product attention[47], formally defined as follows:

**Definition 2.1** (Scaled Dot-Product Attention). Let $\boldsymbol{K} \in \mathbb{R}^{N \times d_k}$ be a *key* matrix with $N$ key vectors, and $\boldsymbol{V} \in \mathbb{R}^{N \times d_v}$ be a *value* matrix with $N$ corresponding value vectors. Given a *query* matrix $\boldsymbol{Q} \in \mathbb{R}^{M \times d_k}$, *Attention* over $(\boldsymbol{K}, \boldsymbol{V})$ is defined as

$$\text{Attention}(\boldsymbol{Q}, \boldsymbol{K}, \boldsymbol{V}) = \text{softmax}(\frac{\boldsymbol{Q}\boldsymbol{K}^\top}{\sqrt{d_k}})\boldsymbol{V} \tag{2}$$

where the softmax function acts on the rows of matrix $\boldsymbol{Q}\boldsymbol{K}^\top \in \mathbb{R}^{M \times N}$.

Vision Transformer (ViT) [8] employs the same attention mechanism within multiple Multi-head Self-Attention (MSA) layers, which is formally defined as follows:

**Definition 2.2** (Multi-head Self-Attention Layer). Let $\boldsymbol{X}^Q, \boldsymbol{X}^K, \boldsymbol{X}^V$ denote the input query, key, and value matrix, respectively, where $\boldsymbol{X}^Q = \boldsymbol{X}^K = \boldsymbol{X}^V = [\boldsymbol{x}_1, ..., \boldsymbol{x}_N]^\top \in \mathbb{R}^{N \times d}$, and $N$ is the length of the input sequence. The output is expressed as

$$\text{MSA}(\boldsymbol{X}^Q, \boldsymbol{X}^K, \boldsymbol{X}^V) := \text{Concat}(\boldsymbol{h}_1, ..., \boldsymbol{h}_m)W^O \in \mathbb{R}^{N \times d}, \tag{3}$$

$$\boldsymbol{h}_i := \text{Attention}(\boldsymbol{X}^Q W_i^Q, \boldsymbol{X}^K W_i^K, \boldsymbol{X}^V W_i^V), \ i \in [m]. \tag{4}$$

where $W^O \in \mathbb{R}^{md_v \times d}, W_i^Q \in \mathbb{R}^{d \times d_k}, W_i^K \in \mathbb{R}^{d \times d_k}$, and $W_i^V \in \mathbb{R}^{d \times d_v}$ are projection matrices, and $m$ is the number of heads in the MSA layer. In ViTs, they use $d_k = d_v = d/m$.

**Prompt-based continual learning.** Prompt-based approaches have emerged as a promising alternative within rehearsal-free continual learning [61, 52, 44]. In vision tasks, prompt-based methods often leverage a pre-trained ViT as a feature extractor $f_\theta$, with its parameters $\theta$ typically frozen. These methods enhance the model by introducing *prompts*, small sets of learnable parameters that influence the operations of the MSA layer [53]. Prompts are strategically injected into the query, key, and value matrices to guide the ViT in learning new tasks. We denote the prompt parameters by $\boldsymbol{p} \in \mathbb{R}^{L_p \times d}$, where $L_p$ is the sequence length and $d$ is the embedding dimension. Previous work [53] outlines two main prompt-based approaches: Prompt Tuning (ProT) [25] and Prefix Tuning (PreT) [26]. While Prompt Tuning directly concatenates the same prompt parameter $\boldsymbol{p}$ to the query, key, and value, prefix tuning divides $\boldsymbol{p}$ into prefixes $\{\boldsymbol{p}^K, \boldsymbol{p}^V\} \in \mathbb{R}^{\frac{L_p}{2} \times d}$ and appends it to the key and value vectors:

$$f_{\text{prompt}}^{\text{Pre}-\text{T}}(\boldsymbol{p}, \boldsymbol{X}^Q, \boldsymbol{X}^K, \boldsymbol{X}^V) := \text{MSA}\left(\boldsymbol{X}^Q, \begin{bmatrix} \boldsymbol{p}^K \\ \boldsymbol{X}^K \end{bmatrix}, \begin{bmatrix} \boldsymbol{p}^V \\ \boldsymbol{X}^V \end{bmatrix}\right) = \text{Concat}(\tilde{\boldsymbol{h}}_1, ..., \tilde{\boldsymbol{h}}_m)W^O \tag{5}$$

Existing prompt-based methods in CL address catastrophic forgetting by creating new adaptive prompts for each new task. During testing, the model chooses suitable prompt combinations to handle unseen data from any encountered task [49]. L2P [54] proposes a shared prompt pool for all tasks, utilizing a query-key mechanism for prompt selection. Instead of using the same prompt pool across tasks, DualPrompt [53] introduces G-Prompt and E-Prompt to capture task-agnostic and task-specific information, respectively. S-Prompt [52] focuses on learning task-specific prompts and employs a ProT strategy similar to L2P. CODA-Prompt [42] expands the prompt pool across tasks and performs a weighted summation of the prompt pool using attention factors. A recent work, HiDe-Prompt [49], achieves state-of-the-art performance by introducing a new hierarchical decomposition of CIL objectives and optimizing each component for better performance.

In this study, we focus on prefix tuning as our primary prompt-based methodology and follow the framework presented in HiDe-Prompt [49]. During training, HiDe-Prompt co-optimizes task-specific prompts $\boldsymbol{p}_t$ and model's output layer parameters $\psi$ for each new task $t$ using the WTP objective. These prompts are stored within a prompt pool $\mathbf{P} = \{\boldsymbol{p}_1, ..., \boldsymbol{p}_T\}$. At test time, a separate lightweight auxiliary output layer $\hat{h}_\omega : \mathbb{R}^D \to \mathbb{R}^T$, trained with the TII objective, takes the uninstructed representation $f_\theta(x)$ of a new data point $\boldsymbol{x}$ as input to infer the task identity, guiding the selection of the most suitable prompt $\boldsymbol{p}_k$ from the prompt pool $\mathbf{P}$. Subsequently, the final prediction is given as $\hat{y} = h_\psi(f_\theta(\boldsymbol{x}, \boldsymbol{p}_k))$. For further details, please refer to Appendix D.

**Mixture of experts (MoE)** extends classical mixture models with an adaptive gating mechanism [19, 21]. An MoE model consists of a group of $N$ expert networks $f_i : \mathbb{R}^d \to \mathbb{R}^{d_v}$, for all $i \in [N]$, and a gate function $G : \mathbb{R}^d \to \mathbb{R}^N$. Given an input $\boldsymbol{h} \in \mathbb{R}^d$, MoE computes a weighted sum of expert outputs $f_i(\boldsymbol{h})$ based on learned score function $s_i : \mathbb{R}^d \to \mathbb{R}$ for each expert:

$$\boldsymbol{y} := \sum_{j=1}^N G(\boldsymbol{h})_j \cdot f_j(\boldsymbol{h}) := \sum_{j=1}^N \frac{\exp(s_j(\boldsymbol{h}))}{\sum_{\ell=1}^N \exp(s_\ell(\boldsymbol{h}))} \cdot f_j(\boldsymbol{h}), \tag{6}$$

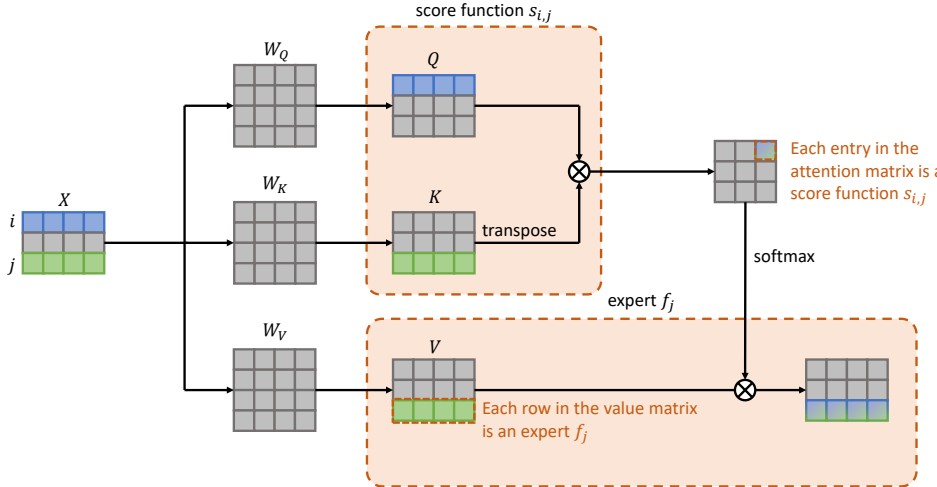

Figure 1: An illustrative depiction of the relationship between self-attention and MoE. Each output vector of a head in the MSA layer can be viewed as the output of a MoE model. These MoE models share the same set of experts encoded in the value matrix. Each entry in the attention matrix corresponds to a score function within this architecture.

where $G(\boldsymbol{h}) := \mathrm{softmax}(s_1(\boldsymbol{h}), \ldots, s_N(h))$. Building on this concept, works by [10, 41] established the MoE layer as a fundamental building block to scale up model capacity efficiently. Please refer to Appendix C for a comprehensive discussion of related works.

## 3 Connection between Prefix Tuning and Mixture of Experts

We first explore the relationship between attention and mixture of experts in Section 3.1, followed by establishing the connection between prefix tuning and the mixture of experts in Section 3.2.

### 3.1 Mixture of Experts Meets Attention

Following the notation established in Definition 2.2, let's consider the $l$-th head within the MSA layer. Let $\boldsymbol{X} = \left[\boldsymbol{x}_1^\top, \ldots, \boldsymbol{x}_N^\top\right]^\top \in \mathbb{R}^{Nd}$, which is the concatenation of input sequence embeddings into a single one-dimensional vector. We define the matrix $E_i \in \mathbb{R}^{d \times Nd}$ such that $E_i \boldsymbol{X} := \boldsymbol{x}_i$ for all $i \in [N]$. Furthermore, we introduce an MoE architecture consisting of a group of $N$ expert networks $f_j : \mathbb{R}^{Nd} \to \mathbb{R}^{d_v}$, $N$ gating functions $G_i : \mathbb{R}^{Nd} \to \mathbb{R}^N$ with the score function for the $j$-th expert of the $i$-th gating $s_{i,j} : \mathbb{R}^{Nd} \to \mathbb{R}$, where

$$f_j(\boldsymbol{X}) := W_l^{V^\top} E_j \boldsymbol{X} = W_l^{V^\top} \boldsymbol{x}_j, \ s_{i,j}(\boldsymbol{X}) := \frac{\boldsymbol{X}^\top E_i^\top W_l^Q W_l^{K^\top} E_j \boldsymbol{X}}{\sqrt{d_v}} = \frac{\boldsymbol{x}_i^\top W_l^Q W_l^{K^\top} \boldsymbol{x}_j}{\sqrt{d_v}}$$

for $i$ and $j \in [N]$. From equation (4), we can express the output of the $l$-th head as follows:

$$\boldsymbol{h}_l = \mathrm{softmax}\left(\frac{\boldsymbol{X}^Q W_l^Q W_l^{K^\top} \boldsymbol{X}^{K^\top}}{\sqrt{d_v}}\right) \boldsymbol{X}^V W_l^V = [\boldsymbol{h}_{l,1}, \ldots, \boldsymbol{h}_{l,N}]^\top \in \mathbb{R}^{N \times d_v}, \tag{7}$$

$$\boldsymbol{h}_{l,i} = \sum_{j=1}^N \frac{\exp\left(\frac{\boldsymbol{x}_i^\top W_l^Q W_l^{K^\top} \boldsymbol{x}_j}{\sqrt{d_v}}\right)}{\sum_{k=1}^N \exp\left(\frac{\boldsymbol{x}_i^\top W_l^Q W_l^{K^\top} \boldsymbol{x}_k}{\sqrt{d_v}}\right)} W_l^{V^\top} \boldsymbol{x}_j = \sum_{j=1}^N \frac{\exp(s_{i,j}(\boldsymbol{X}))}{\sum_{k=1}^N \exp(s_{i,k}(\boldsymbol{X}))} f_j(\boldsymbol{X}), \tag{8}$$

for $i \in [N]$. Expanding on equation (8), we can discern that each attention head within the MSA layer implicitly embodies a special mixture of experts architecture. This architecture encompasses $N$ MoE models, each featuring its own quadratic gating function $G_i$. However, instead of employing $N^2$

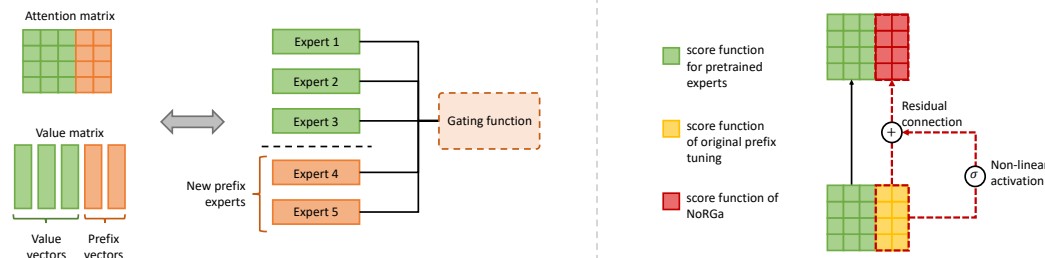

Figure 2: Left: An illustrative depiction of prefix tuning as the introduction of new experts into pre-trained MoE models. Right: Visualization of NoRGa implementation, integrating non-linear activation and residual connections into the prefix tuning attention matrix.

separate expert networks for each model, this architecture utilizes $N$ shared linear expert networks $f_j$ for $j \in [N]$, significantly reducing the number of parameters. Notably, each expert network and its corresponding gating function process the entire input sequence directly, rather than individual embedding $\boldsymbol{x}_i$ as in traditional MoE layers [41]. This connection between self-attention and mixture of experts is depicted in Figure 1. In the subsequent section, we explore how prompt-based techniques can be viewed through this lens.

### 3.2 Prefix Tuning via the Perspective of Mixture of Experts

Building on the connection between self-attention and mixture of experts, we propose that applying prefix tuning can be interpreted as the introduction of new experts to customize the pre-trained model for a specific task, as illustrated in Figure 2. Specifically, similar to Section 3.1, we consider the $l$-th head within the MSA layer. We denote $\boldsymbol{p}^K = \left[\boldsymbol{p}_1^K, \ldots, \boldsymbol{p}_L^K\right]^\top \in \mathbb{R}^{L \times d}$, $\boldsymbol{p}^V = \left[\boldsymbol{p}_1^V, \ldots, \boldsymbol{p}_L^V\right]^\top \in \mathbb{R}^{L \times d}$, where $L = \frac{L_p}{2}$. We define new *prefix* experts $f_{N+j} : \mathbb{R}^{Nd} \to \mathbb{R}^{d_v}$ along with their corresponding new score functions $s_{i,N+j} : \mathbb{R}^{Nd} \to \mathbb{R}$ as follows:

$$f_{N+j}(\boldsymbol{X}) := W_l^{V\top} \boldsymbol{p}_j^V, \quad s_{i,N+j}(\boldsymbol{X}) := \frac{\boldsymbol{X}^\top E_i^\top W_l^Q W_l^{K\top} \boldsymbol{p}_j^K}{\sqrt{d_v}} = \frac{\boldsymbol{x}_i^\top W_l^Q W_l^{K\top} \boldsymbol{p}_j^K}{\sqrt{d_v}} \quad (9)$$

for $i \in [N]$ and $j \in [L]$. Then from equation (5), the output of the $l$-th head can be expressed as:

$$\tilde{\boldsymbol{h}}_l = \text{Attention}\left(\boldsymbol{X}^Q W_l^Q, \begin{bmatrix} \boldsymbol{p}^K \\ \boldsymbol{X}^K \end{bmatrix} W_l^K, \begin{bmatrix} \boldsymbol{p}^V \\ \boldsymbol{X}^V \end{bmatrix} W_l^V\right) = \left[\tilde{\boldsymbol{h}}_{l,1}, \ldots, \tilde{\boldsymbol{h}}_{l,N}\right]^\top \in \mathbb{R}^{N \times d_v}, \quad (10)$$

$$\tilde{\boldsymbol{h}}_{l,i} = \sum_{j=1}^N \frac{\exp(s_{i,j}(\boldsymbol{X}))}{\sum_{k=1}^N \exp(s_{i,k}(\boldsymbol{X})) + \sum_{k'=1}^L \exp(s_{i,N+k'}(\boldsymbol{X}))} f_j(\boldsymbol{X})$$

$$+ \sum_{j'=1}^L \frac{\exp(s_{i,N+j'}(\boldsymbol{X}))}{\sum_{k=1}^N \exp(s_{i,k}(\boldsymbol{X})) + \sum_{k'=1}^L \exp(s_{i,N+k'}(\boldsymbol{X}))} f_{N+j'}(\boldsymbol{X}) \quad (11)$$

It's worth noting that $W_l^Q$, $W_l^K$, and $W_l^V$ remain fixed, with only $\boldsymbol{p}^K$ and $\boldsymbol{p}^V$ being learnable. By examining equation (8) and equation (11), we can interpret each head in a multi-head self-attention layer within a pre-trained model as a mixture of experts architecture with pre-trained experts $f_j$ and gating score functions $s_{i,j}$ for $i$ and $j \in [N]$. Prefix tuning extends this MoE by introducing $L$ additional prefix experts $f_{N+j'}$ defined by prefix vectors $\boldsymbol{p}_{j'}^V$ and linear score functions $s_{i,N+j'}$ for $i \in [N]$ and $j' \in [L]$. These new experts collaborate with the pre-trained ones within the MoE model, facilitating the model's adaptation to downstream tasks.

We argue that our introduction of a novel connection between self-attention, prefix tuning, and MoE offers a fresh perspective on the design of previous prompt-based continual learning methods. In the context of continual learning, the pre-trained experts serve as a knowledge base, while prefix tuning augments it with task-specific knowledge encoded in new experts. Moreover, we draw a parallel between the pre-trained experts and the G(eneral)-Prompt utilized in DualPrompt, which captures

task-agnostic information [53]. Both are shared across tasks, making them useful for prediction, especially when task identification is incorrect. Notably, the new experts achieve their efficiency through simple linear gating functions and independence from the input, unlike the pre-trained experts. For simplicity, we call the MoE model (11) as *linear gating prefix MoE*.

**Statistical suboptimality.** The connection between prefix tuning and the MoE within the linear gating prefix MoE model (11) allows us to theoretically explore the statistical behavior of the prefix tuning. In Appendix A, by interpreting the linear gating prefix MoE as a regression problem with sample size $n$, we demonstrate that the convergence rate for estimating the model parameters, e.g., prompts, can be as slow as $\mathcal{O}(1/\log^\tau(n))$ where $\tau > 0$ is some constant. This suggests that a huge amount of data is required to achieve reasonable parameter estimation in the linear gating prefix MoE model, which can be discouraging in practice. To address this statistical limitation, the next section introduces a novel non-linear residual gating score function to replace the linear gating function.

## 4 Non-linear Residual Gate Meets Prefix Tuning

As discussed earlier, prefix tuning introduces additional experts within the MoE framework, resulting in the linear gating prefix MoE model. However, as outlined in Appendix A, this approach suffers from suboptimal sample efficiency for parameter estimation. To overcome this and enhance overall CIL performance, we propose an innovative approach that significantly improves sample efficiency while promoting WTP performance in Section 4.1 and provide theoretical explanations in Section 4.2.

### 4.1 NoRGa: Non-linear Residual Gate

We propose a simple yet effective modification to the linear gating prefix MoE model by incorporating non-linear activation and residual connection within the score functions of prefix experts as follows:

$$\hat{s}_{i,N+j}(\boldsymbol{X}) := \frac{\boldsymbol{X}^\top E_i^\top W_l^Q W_l^{K^\top} \boldsymbol{p}_j^K}{\sqrt{d_v}} + \alpha \cdot \sigma\left(\tau \cdot \frac{\boldsymbol{X}^\top E_i^\top W_l^Q W_l^{K^\top} \boldsymbol{p}_j^K}{\sqrt{d_v}}\right)$$

$$= s_{i,N+j}(\boldsymbol{X}) + \alpha \cdot \sigma(\tau \cdot s_{i,N+j}(\boldsymbol{X})), \ i \in [N], \ j \in [L], \tag{12}$$

where $\alpha, \tau \in \mathbb{R}$ are learnable scalar factors, and $\sigma$ is a non-linear activation function. The new score function in equation (12) consists of a linear and a non-linear component. We call the new prefix MoE model with score functions (12) as *non-linear residual gating prefix MoE*.

The use of a non-linear activation function here is motivated by the algebraic independence condition in Definition 4.2 to theoretically guarantee the optimal sample efficiency of expert and parameter estimations (cf. Theorem 4.3). It's worth noting that removing the linear component $s_{i,N+j}(\boldsymbol{X})$ in the score function (12) could potentially lead to the vanishing gradient problem during training. To mitigate this challenge, we incorporate a residual connection [14] into the formulation. Our modification introduces minimal additional parameters ($\alpha$ and $\tau$) compared to the original score function, ensuring parameter efficiency. This is particularly crucial in continual learning scenarios where the number of parameters grows with each new task. For implementation, we define $H_l = W_l^Q W_l^{K^\top}$. From equation (5), the attention matrix of the $l$-th head can then be written as:

$$A_l = \frac{\boldsymbol{X}^Q H_l[\boldsymbol{p}^{K^\top}, \ \boldsymbol{X}^{K^\top}]}{\sqrt{d_v}} = \frac{[\boldsymbol{X}^Q H_l \boldsymbol{p}^{K^\top}, \ \boldsymbol{X}^Q H_l \boldsymbol{X}^{K^\top}]}{\sqrt{d_v}} = [A_l^{\text{prompt}}, \ A_l^{\text{pretrain}}]. \tag{13}$$

Here, $A_l^{\text{prompt}}$ denotes the attention score matrix for the prompts, and $A_l^{\text{pretrain}}$ represents the attention score matrix for the pre-trained experts. To implement NoRGa, we can directly modify the final attention matrix as follows:

$$\hat{A}_l = [\hat{A}_l^{\text{prompt}}, A_l^{\text{pretrain}}], \tag{14}$$

$$\hat{A}_l^{\text{prompt}} = A_l^{\text{prompt}} + \alpha \cdot \sigma(\tau \cdot A_l^{\text{prompt}}). \tag{15}$$

The implementation of NoRGa is illustrated in Figure 2. Despite its simplicity, our modification can significantly enhance sample efficiency and promote more reasonable parameter estimation, as demonstrated in our theoretical analysis in Section 4.2. Within the HiDe-Prompt framework, task-specific prompt parameters are trained using the WTP objective for each new task. Consequently, our modification leads to better parameter estimation, which directly contributes to improved WTP performance, ultimately improving overall continual learning efficacy. Importantly, NoRGa maintains the same parameter count as HiDe-Prompt, which is crucial in CL because of the memory constraint. Here, we evaluated $\sigma$ with tanh, sigmoid, and GELU, finding tanh to perform well in most cases.

## 4.2 Theoretical Explanation for Non-linear Residual Gating Prefix MoE

Similar to the setting in Appendix A, we prove that estimating parameters in the non-linear residual gating prefix MoE model (12) is statistically efficient in terms of the number of data. To provide a fair comparison to the linear gating prefix MoE, we focus only on the first head and its first row, namely, $l = 1$ and $i = 1$ in equation (12). Then, we proceed to provide a theoretical justification of our claim by viewing this row as an output of a regression setting. In particular, we assume that $(\boldsymbol{X}_1, Y_1), (\boldsymbol{X}_2, Y_2), \ldots, (\boldsymbol{X}_n, Y_n) \in \mathbb{R}^{Nd} \times \mathbb{R}$ are i.i.d. samples generated from model:

$$Y_i = g_{G_*}(\boldsymbol{X}_i) + \varepsilon_i, \quad i = 1, 2, \ldots, n, \tag{16}$$

where $\varepsilon_1, \ldots, \varepsilon_n$ are independent Gaussian noise variables such that $\mathbb{E}[\varepsilon_i | X_i] = 0$ and $\mathrm{Var}(\varepsilon_i | X_i) = \nu^2$ for all $1 \le i \le n$. Additionally, we assume that $\boldsymbol{X}_1, \boldsymbol{X}_2, \ldots, \boldsymbol{X}_n$ are i.i.d. samples from some probability distribution $\mu$. The regression function $g_{G_*}(\cdot)$ in equation (16) then takes the form of a non-linear residual gating prefix MoE model with $N$ pre-trained experts and $L$ unknown experts,

$$g_{G_*}(\boldsymbol{X}) := \sum_{j=1}^{N} \frac{\exp(\boldsymbol{X}^\top B_j^0 \boldsymbol{X} + c_j^0)}{T(\boldsymbol{X})} \cdot h(\boldsymbol{X}, \eta_j^0)$$

$$+ \sum_{j'=1}^{L} \frac{\exp((\beta_{1j'}^*)^\top \boldsymbol{X} + \alpha \sigma(\tau (\beta_{1j'}^*)^\top \boldsymbol{X}) + \beta_{0j'}^*)}{T(\boldsymbol{X})} \cdot h(\boldsymbol{X}, \eta_{j'}^*), \tag{17}$$

where $T(\boldsymbol{X}) := \sum_{k=1}^{N} \exp(\boldsymbol{X}^\top B_k^0 \boldsymbol{X} + c_k^0) + \sum_{k'=1}^{L} \exp((\beta_{1k'}^*)^\top \boldsymbol{X} + \alpha \sigma(\tau (\beta_{1k'}^*)^\top \boldsymbol{X}) + \beta_{0k'}^*)$, $G_* := \sum_{j'=1}^{L} \exp(\beta_{0j'}^*) \delta_{(\beta_{1j'}^*, \eta_{j'}^*)}$ denotes a *mixing measure,* i.e., a weighted sum of Dirac measures $\delta$, associated with unknown parameters $(\beta_{1j'}^*, \beta_{0j'}^*, \eta_{j'}^*)_{j'=1}^{L}$ in $\mathbb{R}^{Nd} \times \mathbb{R} \times \mathbb{R}^q$. Here, the matrix $B_j^0$ is equivalent to $(E_1^\top W_1^Q W_1^{K^\top} E_j / \sqrt{d_v})$ in the score function $s_{1,j}(\boldsymbol{X})$, and the vector $\beta_{1j'}^*$ corresponds to the vector $(E_1^\top W_1^Q W_1^{K^\top} \boldsymbol{p}_{j'}^K / \sqrt{d_v})$ in $\hat{s}_{1,N+j'}(\boldsymbol{X})$. Furthermore, the experts $h(\boldsymbol{X}, \eta_j^0)$ and $h(\boldsymbol{X}, \eta_{j'}^*)$ represent $f_j(\boldsymbol{X})$ and $f_{N+j'}(\boldsymbol{X})$, respectively. In our formulation, for the generality of the ensuing theory, we consider general parametric forms of the experts $h(\boldsymbol{X}, \eta_j^0)$ and $h(\boldsymbol{X}, \eta_{j'}^*)$, i.e., we do not only constrain these expert functions to be the forms of the simple experts in the aforementioned model. Similar to the setting in Appendix A, $B_j^0$, $c_j^0$, and the expert parameters $\eta_j^0$ are known. Our goal is to estimate the unknown prompt-related parameters $\beta_{1j'}^*$, $\beta_{0j'}^*$, and $\eta_{j'}^*$.

**Least squares estimation.** We will use the least squares method [45] to estimate the unknown parameters $(\beta_{0j'}^*, \beta_{1j'}^*, \eta_{j'}^*)_{j'=1}^{L}$ or, equivalently, the ground-truth mixing measure $G^*$. In particular, we take into account the estimator

$$\widehat{G}_n := \underset{G \in \mathcal{G}_{L'}(\Theta)}{\arg\min} \sum_{i=1}^{n} \left( Y_i - g_G(\boldsymbol{X}_i) \right)^2, \tag{18}$$

where we denote $\mathcal{G}_{L'}(\Theta) := \{ G = \sum_{i=1}^{\ell} \exp(\beta_{0i}) \delta_{(\beta_{1i}, \eta_i)} : 1 \le \ell \le L', (\beta_{0i}, \beta_{1i}, \eta_i) \in \Theta \}$ as the set of all mixing measures with at most $L'$ atoms. In practice, since the true number of experts $L$ is typically unknown, we assume that the number of fitted experts $L'$ is sufficiently large, i.e., $L' > L$.

To begin with, we explore the convergence behavior of the regression estimator $g_{\widehat{G}_n}(\cdot)$ to the true regression function $g_{G_*}(\cdot)$ under the $L_2(\mu)$-norm in the following theorem:

**Theorem 4.1** (Regression Estimation Rate). *Equipped with a least squares estimator $\widehat{G}_n$ given in equation* (18)*, the model estimation $g_{\widehat{G}_n}(\cdot)$ converges to the true model $g_{G_*}(\cdot)$ at the following rate:*

$$\|g_{\widehat{G}_n} - g_{G_*}\|_{L_2(\mu)} = \mathcal{O}_P(\sqrt{\log(n)/n}). \tag{19}$$

Proof of Theorem 4.1 is in Appendix B.1. The bound (19) implies that the rate for estimating the regression function $g_{G_*}(\cdot)$ is of order $\mathcal{O}_P(\sqrt{\log(n)/n})$, which is parametric on the sample size $n$. More importantly, it also indicates that if there exists a loss function among parameters $\mathcal{L}$ such that $\|g_{\widehat{G}_n} - g_{G_*}\|_{L_2(\mu)} \gtrsim \mathcal{L}(\widehat{G}_n, G_*)$, then we would obtain the bound $\mathcal{L}(\widehat{G}_n, G_*) = \mathcal{O}_P(\sqrt{\log(n)/n})$, which leads to the desired parameter and expert estimation rates.

We now turn our attention to the parameter and expert estimation problems. To understand how the non-linear residual gating affects these problems, we analyze the properties of the expert $h(\cdot, \eta)$ and the activation function $\sigma(\cdot)$ to determine which formulations will achieve favorable performance.

**Definition 4.2** (Algebraic independence). We say that an expert function $h(\cdot, \eta)$ and an activation function $\sigma(\cdot)$ are algebraically independent if they are twice differentiable w.r.t their parameters, and if for any $k \geq 1$ and pair-wise distinct parameters $(\beta_{11}, \eta_1), \ldots, (\beta_{1k}, \eta_k)$, the following set of functions in $\boldsymbol{X}$ is linearly independent for almost every $\boldsymbol{X} \in \mathbb{R}^{Nd}$:

$$\Big\{ \boldsymbol{X}^\nu \Big[ (1 + \sigma'(\beta_{1j}^\top \boldsymbol{X}))^{|\nu|} + \mathbf{1}_{\{|\nu|=2\}} \sigma''(\beta_{1j}^\top \boldsymbol{X}) \Big] \cdot \frac{\partial^{|\gamma|} h}{\partial \eta^\gamma}(\boldsymbol{X}, \eta_j) : j \in [k_*],$$

$$\nu \in \mathbb{N}^{Nd}, \gamma \in \mathbb{N}^q : 0 \leq |\nu| + |\gamma| \leq 2 \Big\}.$$

Intuitively, the algebraic independence condition ensures that there will be no interactions among parameters of the expert function $h(\cdot, \eta)$ and the activation function $\sigma(\cdot)$. Technically, a key step in our argument is to decompose the regression discrepancy $g_{\widehat{G}_n}(\boldsymbol{X}) - g_{G_*}(\boldsymbol{X})$ into a combination of linearly independent terms by applying Taylor expansions to the product of the softmax's numerator and the expert function, i.e., $\exp(\beta_1^\top \boldsymbol{X} + \alpha \sigma(\tau \beta_1^\top \boldsymbol{X})) h(\boldsymbol{X}, \eta)$. Thus, the above condition guarantees that all the derivative terms in the Taylor expansion are linearly independent. To exemplify the algebraic independence condition, we consider the following simple examples of the expert functions $h(\cdot, \eta)$ and the activation $\sigma(\cdot)$ that are algebraically independent.

**Example.** When the expert function $h(\cdot, \eta)$ is formulated as a neural network $h(\boldsymbol{X}, (a, b)) = \varphi(a^\top \boldsymbol{X} + b)$ with the activation $\varphi(\cdot) \in \{\text{ReLU}(\cdot), \text{GELU}(\cdot), z \mapsto z^p\}$, where $(a, b) \in \mathbb{R}^{Nd} \times \mathbb{R}$, and the activation function $\sigma(\cdot)$ is one among the functions $\text{sigmoid}(\cdot), \tanh(\cdot), \text{GELU}(\cdot)$, then they satisfy the algebraic independence condition in Definition 4.2.

Finally, we establish the rates for estimating parameters and experts in the non-linear residual gating prefix MoE model in Theorem 4.3. Prior to presenting the theorem statement, let us design a loss function among parameters based on a notion of Voronoi cells [27], which is a commonly employed approach for the convergence analysis of expert estimation in MoE models [36, 34, 35, 33], yet tailored to the setting of this paper. In particular, the Voronoi loss used for our analysis is defined as

$$\mathcal{L}_1(G, G_*) := \sum_{j' \in [L]:|\mathcal{V}_{j'}|>1} \sum_{i \in \mathcal{V}_{j'}} \exp(\beta_{0i}) \Big[ \|\Delta\beta_{1ij'}\|^2 + \|\Delta\eta_{ij'}\|^2 \Big]$$

$$+ \sum_{j' \in [L]:|\mathcal{V}_{j'}|=1} \sum_{i \in \mathcal{V}_{j'}} \exp(\beta_{0i}) \Big[ \|\Delta\beta_{1ij'}\| + \|\Delta\eta_{ij'}\| \Big] + \sum_{j'=1}^{L} \Big| \sum_{i \in \mathcal{V}_{j'}} \exp(\beta_{0i}) - \exp(\beta_{0j'}^*) \Big|, \quad (20)$$

where we denote $\Delta\beta_{1ij'} := \beta_{1i} - \beta_{1j'}^*$ and $\Delta\eta_{ij'} := \eta_i - \eta_{j'}^*$. Above, $\mathcal{V}_{j'} \equiv \mathcal{V}_{j'}(G)$, for $j' \in [L]$, is a Voronoi cell associated with the mixing measure $G$ generated by the true component $\omega_j^* := (\beta_{1j'}^*, \eta_{j'}^*)$, which is defined as follows:

$$\mathcal{V}_{j'} := \{i \in \{1, 2, \ldots, L'\} : \|\omega_i - \omega_{j'}^*\| \leq \|\omega_i - \omega_\ell^*\|, \forall \ell \neq j'\}, \quad (21)$$

where we denote $\omega_i := (\beta_{1i}, \eta_i)$ as a component of $G$. Note that, the cardinality of each Voronoi cell $\mathcal{V}_{j'}$ indicates the number of components $\omega_i$ of $G$ approximating the true component $\omega_{j'}^*$ of $G_*$. Additionally, since $\mathcal{L}_1(G, G_*) = 0$ if and only if $G \equiv G_*$, it follows that when $\mathcal{L}_1(G, G_*)$ becomes sufficiently small, the differences $\Delta\beta_{1ij'}$ and $\Delta\eta_{ij'}$ are also small. This observation indicates that, although $\mathcal{L}_1(G, G_*)$ is a proper metric as it is not symmetric, it is an appropriate loss function for measuring the discrepancy between the least square estimator $\widehat{G}_n$ and the true mixing measures $G_*$.

**Theorem 4.3.** *Assume that the expert function $h(x, \eta)$ and the activation $\sigma(\cdot)$ are algebraically independent, then we achieve the following lower bound for any $G \in \mathcal{G}_{L'}(\Theta)$:*

$$\|g_G - g_{G_*}\|_{L_2(\mu)} \gtrsim \mathcal{L}_1(G, G_*),$$

*which together with Theorem 4.1 indicates that $\mathcal{L}_1(\widehat{G}_n, G_*) = \widetilde{\mathcal{O}}_P(n^{-1/2})$.*

Proof of Theorem 4.3 is in Appendix B.2. A few comments on Theorem 4.3 are in order: (i) From the bound $\mathcal{L}_1(\widehat{G}_n, G_*) = \widetilde{\mathcal{O}}_P(n^{-1/2})$, we deduce that the estimation rates for the over-specified parameters $\beta_{1j'}^*, \eta_{1j'}^*$, where $j' \in [L] : |\mathcal{V}_{j'}| > 1$, are all of order $\mathcal{O}_P(\sqrt[4]{\log(n)/n})$. Since the expert $h(\cdot, \eta)$ is twice differentiable over a bounded domain, it is also a Lipschitz function. Thus, denote

Table 1: Overall performance comparison on Split CIFAR-100 and Split ImageNet-R. We present Final Average Accuracy (FA), Cumulative Average Accuracy (CA), and Average Forgetting Measure (FM) of all methods under different pre-trained models.

| PTM | Method | Split CIFAR-100 | | | Split Imagenet-R | | |
|---|---|---|---|---|---|---|---|
| | | FA ($\uparrow$) | CA($\uparrow$) | FM($\downarrow$) | FA ($\uparrow$) | CA($\uparrow$) | FM($\downarrow$) |
| Sup-21K | L2P | $83.06 \pm 0.17$ | $88.27 \pm 0.71$ | $5.61 \pm 0.32$ | $67.53 \pm 0.44$ | $71.98 \pm 0.52$ | $5.84 \pm 0.38$ |
| | DualPrompt | $87.30 \pm 0.27$ | $91.23 \pm 0.65$ | $3.87 \pm 0.43$ | $70.93 \pm 0.08$ | $75.67 \pm 0.52$ | $5.47 \pm 0.19$ |
| | S-Prompt | $87.57 \pm 0.42$ | $91.38 \pm 0.69$ | $3.63 \pm 0.41$ | $69.88 \pm 0.51$ | $74.25 \pm 0.55$ | $4.73 \pm 0.47$ |
| | CODA-Prompt | $86.94 \pm 0.63$ | $91.57 \pm 0.75$ | $4.04 \pm 0.18$ | $70.03 \pm 0.47$ | $74.26 \pm 0.24$ | $5.17 \pm 0.22$ |
| | HiDe-Prompt | $92.61 \pm 0.28$ | $94.03 \pm 0.01$ | $1.50 \pm 0.28$ | $75.06 \pm 0.12$ | $76.60 \pm 0.01$ | $\mathbf{4.09} \pm 0.13$ |
| | NoRGa (Ours) | $\mathbf{94.48} \pm 0.13$ | $\mathbf{95.83} \pm 0.37$ | $\mathbf{1.44} \pm 0.27$ | $\mathbf{75.40} \pm 0.39$ | $\mathbf{79.52} \pm 0.07$ | $4.59 \pm 0.07$ |
| iBOT-21K | L2P | $79.13 \pm 1.25$ | $85.13 \pm 0.05$ | $7.50 \pm 1.21$ | $61.31 \pm 0.50$ | $68.81 \pm 0.52$ | $10.72 \pm 0.40$ |
| | DualPrompt | $78.84 \pm 0.47$ | $86.16 \pm 0.02$ | $8.84 \pm 0.67$ | $58.69 \pm 0.61$ | $66.61 \pm 0.67$ | $11.75 \pm 0.92$ |
| | S-Prompt | $79.14 \pm 0.65$ | $85.85 \pm 0.17$ | $8.23 \pm 1.15$ | $57.96 \pm 1.10$ | $66.42 \pm 0.71$ | $11.27 \pm 0.72$ |
| | CODA-Prompt | $80.83 \pm 0.27$ | $87.02 \pm 0.20$ | $7.50 \pm 0.25$ | $61.22 \pm 0.35$ | $66.76 \pm 0.37$ | $9.66 \pm 0.20$ |
| | HiDe-Prompt | $93.02 \pm 0.15$ | $94.56 \pm 0.05$ | $\mathbf{1.26} \pm 0.13$ | $70.83 \pm 0.17$ | $73.23 \pm 0.08$ | $\mathbf{6.77} \pm 0.23$ |
| | NoRGa (Ours) | $\mathbf{94.76} \pm 0.15$ | $\mathbf{95.86} \pm 0.31$ | $1.34 \pm 0.14$ | $\mathbf{73.06} \pm 0.26$ | $\mathbf{77.46} \pm 0.42$ | $6.88 \pm 0.49$ |
| iBOT-1K | L2P | $75.51 \pm 0.88$ | $82.53 \pm 1.10$ | $6.80 \pm 1.70$ | $59.43 \pm 0.28$ | $66.83 \pm 0.92$ | $11.33 \pm 1.25$ |
| | DualPrompt | $76.21 \pm 1.00$ | $83.54 \pm 1.23$ | $9.89 \pm 1.81$ | $60.41 \pm 0.76$ | $66.87 \pm 0.41$ | $9.21 \pm 0.43$ |
| | S-Prompt | $76.60 \pm 0.61$ | $82.89 \pm 0.89$ | $8.60 \pm 1.36$ | $59.56 \pm 0.60$ | $66.60 \pm 0.13$ | $8.83 \pm 0.81$ |
| | CODA-Prompt | $79.11 \pm 1.02$ | $86.21 \pm 0.49$ | $7.69 \pm 1.57$ | $66.56 \pm 0.68$ | $73.14 \pm 0.57$ | $7.22 \pm 0.38$ |
| | HiDe-Prompt | $93.48 \pm 0.11$ | $95.02 \pm 0.01$ | $\mathbf{1.63} \pm 0.10$ | $71.33 \pm 0.21$ | $73.62 \pm 0.13$ | $7.11 \pm 0.02$ |
| | NoRGa (Ours) | $\mathbf{94.01} \pm 0.04$ | $\mathbf{95.11} \pm 0.35$ | $1.61 \pm 0.30$ | $\mathbf{72.77} \pm 0.20$ | $\mathbf{76.55} \pm 0.46$ | $\mathbf{7.10} \pm 0.39$ |
| DINO-1K | L2P | $72.23 \pm 0.35$ | $79.71 \pm 1.26$ | $8.37 \pm 2.30$ | $57.21 \pm 0.69$ | $64.09 \pm 0.74$ | $7.47 \pm 0.96$ |
| | DualPrompt | $73.95 \pm 0.49$ | $81.85 \pm 0.59$ | $9.32 \pm 1.42$ | $57.98 \pm 0.71$ | $65.39 \pm 0.27$ | $9.32 \pm 0.69$ |
| | S-Prompt | $74.39 \pm 0.17$ | $81.60 \pm 0.74$ | $9.07 \pm 1.13$ | $57.55 \pm 0.72$ | $64.90 \pm 0.13$ | $8.73 \pm 0.56$ |
| | CODA-Prompt | $77.50 \pm 0.64$ | $84.81 \pm 0.30$ | $8.10 \pm 0.01$ | $63.15 \pm 0.39$ | $69.73 \pm 0.25$ | $6.86 \pm 0.11$ |
| | HiDe-Prompt | $92.51 \pm 0.11$ | $94.25 \pm 0.01$ | $1.67 \pm 0.20$ | $68.11 \pm 0.18$ | $71.70 \pm 0.01$ | $6.45 \pm 0.58$ |
| | NoRGa (Ours) | $\mathbf{93.43} \pm 0.33$ | $\mathbf{94.65} \pm 0.62$ | $\mathbf{1.65} \pm 0.25$ | $\mathbf{71.77} \pm 0.44$ | $\mathbf{75.76} \pm 0.49$ | $\mathbf{6.42} \pm 0.68$ |
| MoCo-1K | L2P | $77.24 \pm 0.69$ | $83.73 \pm 0.70$ | $5.57 \pm 0.75$ | $54.13 \pm 0.67$ | $62.09 \pm 0.76$ | $\mathbf{4.88} \pm 0.42$ |
| | DualPrompt | $77.56 \pm 0.63$ | $84.37 \pm 0.51$ | $6.54 \pm 0.50$ | $54.45 \pm 0.30$ | $62.92 \pm 0.41$ | $5.34 \pm 0.41$ |
| | S-Prompt | $77.20 \pm 0.39$ | $84.47 \pm 0.37$ | $7.00 \pm 0.62$ | $53.94 \pm 0.32$ | $62.42 \pm 0.51$ | $5.16 \pm 0.48$ |
| | CODA-Prompt | $77.83 \pm 0.34$ | $84.97 \pm 0.23$ | $12.60 \pm 0.02$ | $55.75 \pm 0.26$ | $65.49 \pm 0.36$ | $10.46 \pm 0.04$ |
| | HiDe-Prompt | $91.57 \pm 0.20$ | $93.70 \pm 0.01$ | $\mathbf{1.51} \pm 0.17$ | $63.77 \pm 0.49$ | $68.26 \pm 0.01$ | $9.37 \pm 0.71$ |
| | NoRGa (Ours) | $\mathbf{93.52} \pm 0.06$ | $\mathbf{94.94} \pm 0.29$ | $1.63 \pm 0.13$ | $\mathbf{64.52} \pm 0.16$ | $\mathbf{70.21} \pm 0.64$ | $9.06 \pm 0.19$ |

$\widehat{G}_n := \sum_{i=1}^{L_n} \exp(\widehat{\beta}_{0i}) \delta_{(\widehat{\beta}_{1i}^n, \widehat{\eta}_i^n)}$, we achieve that

$$\sup_{\boldsymbol{X}} |h(\boldsymbol{X}, \widehat{\eta}_i^n) - h(\boldsymbol{X}, \eta_{j'}^*)| \lesssim \|\widehat{\eta}_i^n - \eta_{j'}^*\| \lesssim \mathcal{O}_P(\sqrt[4]{\log(n)/n}). \tag{22}$$

The above bound indicates that if the experts $h(\cdot, \eta_j^*)$ are fitted by at least two other experts, then their estimation rates are also of order $\mathcal{O}_P(\sqrt[4]{\log(n)/n})$; (ii) For exactly-specified parameters $\beta_{1j'}^*, \eta_{j'}^*$, where $j' \in [L] : |\mathcal{V}_{j'}| = 1$, the rates for estimating them are faster than those of their over-specified counterparts, standing at order $\mathcal{O}_P(\sqrt{\log(n)/n})$. By arguing similarly to equation (22), the experts $h(\cdot, \eta_{j'}^*)$ also enjoy the faster estimation rate of order $\mathcal{O}_P(\sqrt{\log(n)/n})$, which is parametric on the sample size $n$; (iii) It follows from the above rates that we only need a polynomial number of data (roughly $\epsilon^{-4}$ where $\epsilon$ is the desired approximation error) to estimate the parameters and experts of the non-linear residual gating prefix MoE. By contrast, when using the linear gating, as being demonstrated in Appendix A, it requires an exponential number of data. This highlights the statistical benefits of using the non-linear residual gating MoE model over the linear gating prefix MoE model.

## 5   Experiments

**Datasets.** We evaluate various continual learning methods on widely used CIL benchmarks, including Split CIFAR-100 [23] and Split ImageNet-R [23], consistent with prior work [49]. We further explore the model's performance on fine-grained classification tasks with Split CUB-200 [48] and large inter-task differences with 5-Datasets [9]. Please refer to Appendix E for more details.

**Evaluation Metrics.** We utilize several established metrics described in [50]. These include: final average accuracy (FA), which represents the average accuracy after the final task; cumulative average accuracy (CA), which refers to the historical average accuracy; and average forgetting measure (FM). We give more emphasis to FA and CA due to their comprehensiveness, as noted in [42].

**Baselines.** We compare our approach with several representative prompt-based approaches including L2P [54], DualPrompt [53], CODA-Prompt [42], S-Prompt [52], and HiDe-Prompt [49]. Additionally,

Table 2: Final average accuracy (FA) on Split CUB-200 and 5-Datasets.

| Method | Split CUB-200 | | 5-Datasets | |
| --- | --- | --- | --- | --- |
| | Sup-21K | iBOT-21K | Sup-21K | iBOT-21K |
| L2P | 75.46 | 46.60 | 81.84 | 82.25 |
| DualPrompt | 77.56 | 45.93 | 77.91 | 68.03 |
| S-Prompt | 77.13 | 44.22 | 86.06 | 77.20 |
| CODA-Prompt | 74.34 | 47.79 | 64.18 | 51.65 |
| HiDe-Prompt | 86.56 | 78.23 | 93.83 | 94.88 |
| NoRGa (Ours) | **90.90** | **80.69** | **94.16** | **94.92** |

Table 3: Ablation study of different activation functions, measured by final average accuracy (FA).

| Method | Split CIFAR-100 | | Split CUB-200 | |
| --- | --- | --- | --- | --- |
| | Sup-21K | iBOT-21K | Sup-21K | iBOT-21K |
| HiDe-Prompt | 92.61 | 93.02 | 86.56 | 78.23 |
| NoRGa $\tanh$ | 94.36 | **94.76** | 90.87 | **80.69** |
| NoRGa $\mathrm{sigmoid}$ | **94.48** | 94.69 | **90.90** | 80.18 |
| NoRGa GELU | 94.05 | 94.63 | 90.74 | 80.54 |

we evaluate against state-of-the-art pre-trained model-based continual learning methods, including ADAM [60] and RanPAC [29]. We further extend our evaluation by applying HiDe-Prompt with parameter-efficient fine-tuning techniques like LoRA [17] and Adapters [16]. In line with [49], we utilize the checkpoints of ViT that use supervised pre-training of Imagenet-21K (denoted as Sup-21K), and some self-supervised pre-training such as iBOT-21K, iBOT-1K [62], DINO-1K [4], and MoCo-1K [7]. For implementation details, see Appendix E.

**Main Results.** In Table 1, we evaluate several continual learning methods on Split CIFAR-100 and Split ImageNet-R using diverse pre-trained models. NoRGa achieves state-of-the-art FA and CA across all datasets and models, consistently outperforming HiDe-Prompt. On Sup-21K, NoRGa demonstrates impressive FA results on both CIFAR-100 and ImageNet-R. It also maintains the highest CA, with significant margins of 1.80% and 2.92% on CIFAR-100 and ImageNet-R, respectively, compared to HiDe-Prompt. These results highlight NoRGa's strong ability to retain knowledge and exhibit minimal forgetting, as evidenced by the low FM values on both datasets. NoRGa also surpasses HiDe-Prompt on self-supervised models, with FA improvements up to 1.95% and 3.66%. We further investigate two scenarios: fine-grained classification tasks and large inter-task differences through experiments on Split CUB-200 and 5-Datasets, respectively, as shown in Table 2. NoRGa maintains its lead, achieving FA gaps of 4.34% and 2.46% on Split CUB-200, and the highest FA on 5-Datasets. While gains in some metrics may be modest, NoRGa consistently outperforms HiDe-Prompt in either FA or CA, underscoring its robustness. For example, on Split ImageNet-R with Sup-21K weights, the FA improvement is small (75.06% vs. 75.40%), but the CA gains are substantial (76.60% vs. 79.52%), demonstrating the method's effectiveness and robustness.

**Ablation Study.** To assess the impact of non-linear activation functions on NoRGa's performance, we evaluated the model's behavior with different choices for the activation function $\sigma$, including $\tanh$, $\mathrm{sigmoid}$, and GELU in Table 3. The results show that NoRGa achieves state-of-the-art performance on both Split CIFAR-100 and Split CUB-200 datasets with all three activation functions. These findings suggest that NoRGa exhibits robustness to the choice of non-linear activation within a reasonable range. While all functions perform well, the $\tanh$ activation function demonstrates generally strong performance across scenarios. Further results are provided in the Appendix.

## 6 Conclusion

This paper presents an initial exploration of self-attention and prefix-tuning through the lens of mixture of experts. We find that applying prefix tuning can be viewed as introducing new prefix experts to adapt the pre-trained model. However, limitations in sample efficiency exist. We address this by proposing NoRGa, a novel gating mechanism to enhance continual learning performance. Our results demonstrate NoRGa's effectiveness both theoretically and empirically. While the current implementation of the expert network prioritizes simplicity, future research directions could involve investigating more intricate architectures. Furthermore, the choice of activation functions in our work requires fine-tuning, which opens avenues for future research on adaptively learning activation.

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

# Supplement to "Mixture of Experts Meets Prompt-Based Continual Learning"

In this supplementary material, we first analyze the statistical suboptimality of the Linear Gating Prefix MoE Model (11) in Appendix A. Appendix B provides proofs for the theoretical results presented in Section 4.2. In Appendix C, we discuss related works on mixture of experts. Appendix D outlines the training algorithm for HiDe-Prompt while Appendix E presents the experimental setup and details. Further, Appendix F presents further experiments on the task-incremental learning setting to empirically demonstrate the benefits of using our proposed Non-linear Residual Gating Prefix MoE (12) over the Linear Gating Prefix MoE Model. Appendix G and Appendix H compare NoRGa with other parameter-efficient fine-tuning techniques and pre-trained model-based methods. In Appendix I, we present the efficiency tests, while Appendix J explores the impact of learnable $\alpha$ and $\tau$. Finally, Appendix K compares the training times between NoRGa and HiDe-Prompt.

## A    Statistical Suboptimality of Linear Gating Prefix MoE Model

In this appendix, we demonstrate that estimating parameters and experts in the linear gating prefix MoE model (11) can be statistically inefficient in terms of the number of data. To simplify our findings, we particularly focus on the first head, namely, $l = 1$ in equation (11), and the first row of this head, namely, $i = 1$ in equation (11). Then, we proceed to provide a theoretical justification of our claim for the suboptimality of the linear gating prefix MoE by viewing this row as an output of the regression setting. In particular, we assume that $(\boldsymbol{X}_1, Y_1), (\boldsymbol{X}_2, Y_2), \ldots, (\boldsymbol{X}_n, Y_n) \in \mathbb{R}^{Nd} \times \mathbb{R}$ is an i.i.d. sample generated from the following model:

$$Y_i = f_{G_*}(\boldsymbol{X}_i) + \varepsilon_i, \quad i = 1, 2, \ldots, n, \tag{23}$$

where $\varepsilon_1, \ldots, \varepsilon_n$ are independent Gaussian noise variables such that $\mathbb{E}[\varepsilon_i | \boldsymbol{X}_i] = 0$ and $\mathrm{Var}(\varepsilon_i | \boldsymbol{X}_i) = \nu^2$ for all $1 \leq i \leq n$. Additionally, we assume that $\boldsymbol{X}_1, \boldsymbol{X}_2, \ldots, \boldsymbol{X}_n$ are i.i.d. samples from some probability distribution $\mu$. Motivated by linear gating prefix MoE model (11), the regression function $f_{G_*}(\cdot)$ in equation (23) admits the form of the linear gating prefix MoE model with pre-trained $N$ experts and $L$ unknown experts, namely

$$f_{G_*}(\boldsymbol{X}) := \sum_{j=1}^{N} \frac{\exp(\boldsymbol{X}^\top B_j^0 \boldsymbol{X} + c_j^0)}{\sum_{k=1}^{N} \exp(\boldsymbol{X}^\top B_k^0 \boldsymbol{X} + c_k^0) + \sum_{k'=1}^{L} \exp((\beta_{1k'}^*)^\top \boldsymbol{X} + \beta_{0k'}^*)} \cdot h(\boldsymbol{X}, \eta_j^0)$$

$$+ \sum_{j'=1}^{L} \frac{\exp((\beta_{1j'}^*)^\top \boldsymbol{X} + \beta_{0j'}^*)}{\sum_{k=1}^{N} \exp(\boldsymbol{X}^\top B_k^0 \boldsymbol{X} + c_k^0) + \sum_{k'=1}^{L} \exp((\beta_{1k'}^*)^\top \boldsymbol{X} + \beta_{0k'}^*)} \cdot h(\boldsymbol{X}, \eta_{j'}^*), \tag{24}$$

where $G_* := \sum_{j'=1}^{L} \exp(\beta_{0j'}^*) \delta_{(\beta_{1j'}^*, \eta_{j'}^*)}$ denotes a *mixing measure,* i.e., a weighted sum of Dirac measures $\delta$, associated with unknown parameters $(\beta_{1j'}^*, \beta_{0j'}^*, \eta_{j'}^*)_{j'=1}^{L}$ in $\mathbb{R}^{Nd} \times \mathbb{R} \times \mathbb{R}^q$. Here, the matrix $B_j^0$ plays the role of the matrix $\frac{E_1^\top W_1^Q W_1^{K^\top} E_j}{\sqrt{d_v}}$ in the score function $s_{1,j}(\boldsymbol{X})$. Furthermore, the vector $\beta_{1j'}^*$ corresponds to the vector $\frac{E_i^\top W_l^Q W_l^{K^\top} \boldsymbol{p}_{j'}^K}{\sqrt{d_v}}$ in the score function $s_{1,N+j'}(\boldsymbol{X})$. Furthermore, the experts $h(\boldsymbol{X}, \eta_j^0)$ correspond to the role of $f_j(\boldsymbol{X})$ and $h(\boldsymbol{X}, \eta_{j'}^*)$ correspond to the role of $f_{N+j'}(\boldsymbol{X})$. In our formulation, we consider general parametric forms of the experts $h(\boldsymbol{X}, \eta_j^0)$ and $h(\boldsymbol{X}, \eta_{j'}^*)$, i.e., we do not only constrain these expert functions to be the forms of the simple experts in the linear gating prefix MoE model.

Similar to the linear gating prefix MoE model (11), the matrices $B_j^0$, the biases $c_j^0$, and the expert parameters $\eta_j^0$ are known. Our aim is to estimate the unknown gating parameters $\beta_{1j'}^*, \beta_{0j'}^*$, and $\eta_{j'}^*$ that correspond to the prompts.

**Least squares estimation:** We will use the least squares method [45] to estimate the unknown parameters $(\beta_{0j'}^*, \beta_{1j'}^*, \eta_{j'}^*)_{j'=1}^{L}$ or, equivalently, the ground-truth mixing measure $G^*$. In particular, we take into account the estimator

$$\widetilde{G}_n := \underset{G \in \mathcal{G}_{L'}(\Theta)}{\arg\min} \sum_{i=1}^{n} \left( Y_i - f_G(\boldsymbol{X}_i) \right)^2, \tag{25}$$

where we denote $\mathcal{G}_{L'}(\Theta) := \{G = \sum_{i=1}^{\ell} \exp(\beta_{0i}) \delta_{(\beta_{1i}, \eta_i)} : 1 \leq \ell \leq L', \ (\beta_{0i}, \beta_{1i}, \eta_i) \in \Theta\}$ as the set of all mixing measures with at most $L'$ atoms. In practice, since the true number of true experts $L$ is typically unknown, we assume that the number of fitted experts $L'$ is sufficiently large, i.e. $L' > L$.

Let us recall that our main objective in this appendix is to show that using the linear gating in the prefix MoE model is not sample efficient. To illustrate that point, we consider a simple scenario when the expert function takes the form $h(\boldsymbol{X}, (a, b)) = (a^\top \boldsymbol{X} + b)^p$, for some $p \in \mathbb{N}$. Additionally, we also design a new Voronoi loss function as below to facilitate our arguments.

$$\mathcal{L}_{2,r}(G, G_*) := \sum_{j=1}^{L} \Big| \sum_{i \in \mathcal{V}_j} \exp(\beta_{0i}) - \exp(\beta_{0j}^*) \Big| + \sum_{j=1}^{L} \sum_{i \in \mathcal{V}_j} \exp(\beta_{0i}) \Big[ \|\Delta\beta_{1ij}\|^r + \|\Delta a_{ij}\|^r + |\Delta b_{ij}|^r \Big], \tag{26}$$

where we denote $\Delta\beta_{1ij'} := \beta_{1i} - \beta_{1j'}^*$ and $\Delta\eta_{ij'} := \eta_i - \eta_{j'}^*$.

Now, we are ready to state the result of parameter estimation under the linear gating prefix MoE model in the following theorem:

**Theorem A.1.** *Assume that the experts take the form $h(\boldsymbol{X}, (a, b)) = (a^\top \boldsymbol{X} + b)^p$, for some $p \in \mathbb{N}$, then we achieve the following minimax lower bound of estimating $G_*$:*

$$\inf_{\overline{G}_n \in \mathcal{G}_{L'}(\Theta)} \sup_{G \in \mathcal{G}_{L'}(\Theta) \backslash \mathcal{G}_{L-1}(\Theta)} \mathbb{E}_{f_G}[\mathcal{L}_{2,r}(\overline{G}_n, G)] \gtrsim n^{-1/2},$$

*for any $r \geq 1$, where $\mathbb{E}_{f_G}$ indicates the expectation taken w.r.t the product measure with $f_G^n$.*

There are two main implications of the result in Theorem A.1:

**(i)** The rates for estimating parameters $\beta_{1j}^*$, $a_j^*$ and $b_j^*$ are slower than $\mathcal{O}_P(n^{-1/2r})$, for any $r \geq 1$. This means that they are slower than any polynomial rates, and could be of order $\mathcal{O}_P(1/\log(n))$. Using the same reasoning described after equation (22), we have

$$\sup_x |\varphi((\widehat{a}_i^n)^\top \boldsymbol{X} + \widehat{b}_i^n) - \varphi((a_j^*)^\top \boldsymbol{X} + b_j^*)| \lesssim \cdot \|\widehat{a}_i^n - a_j^*\| + |\widehat{b}_i^n - b_j^*|. \tag{27}$$

As a consequence, the rates for estimating experts $\varphi((a_j^*)^\top \boldsymbol{X} + b_j^*)$ are no better than those for estimating the parameters $a_j^*$ and $b_j^*$, and could also be as slow as $\mathcal{O}_P(1/\log(n))$.

**(ii)** The above rates imply that we need an exponential number of data (roughly $\exp(1/\epsilon^\tau)$ where $\epsilon$ is the desired approximation error) to estimate the parameters and experts of the linear gating prefix MoE. This fact demonstrates that using the linear gating in the prefix MoE model is not sample efficient from the perspective of the expert estimation problem.

*Proof of Theorem A.1.* Prior to presenting the main proof of Proposition A.1, let us introduce the following key result:

**Lemma A.2.** *If the following holds for any $r \geq 1$:*

$$\lim_{\varepsilon \to 0} \inf_{G \in \mathcal{G}_{L'}(\Theta) : \mathcal{L}_{2,r}(G, G_*) \leq \varepsilon} \frac{\|f_G - f_{G_*}\|_{L_2(\mu)}}{\mathcal{L}_{2,r}(G, G_*)} = 0, \tag{28}$$

*then we obtain that*

$$\inf_{\overline{G}_n \in \mathcal{G}_{L'}(\Theta)} \sup_{G \in \mathcal{G}_{L'}(\Theta) \backslash \mathcal{G}_{L-1}(\Theta)} \mathbb{E}_{f_G}[\mathcal{L}_{2,r}(\overline{G}_n, G)] \gtrsim n^{-1/2}. \tag{29}$$

*Proof of Lemma A.2.* Indeed, from the Gaussian assumption on the noise variables $\epsilon_i$, we obtain that $Y_i | \boldsymbol{X}_i \sim \mathcal{N}(f_{G_*}(\boldsymbol{X}_i), \sigma^2)$ for all $i \in [n]$. Next, the assumption in equation (28) indicates for sufficiently small $\varepsilon > 0$ and a fixed constant $C_1 > 0$ which we will choose later, we can find a mixing measure $G_*' \in \mathcal{G}_{L'}(\Theta)$ such that $\mathcal{L}_{2,r}(G_*', G_*) = 2\varepsilon$ and $\|f_{G_*'} - f_{G_*}\|_{L^2(\mu)} \leq C_1\varepsilon$. From Le Cam's

lemma [57], as the Voronoi loss function $\mathcal{L}_{2,r}$ satisfies the weak triangle inequality, we obtain that

$$\inf_{\overline{G}_n\in\mathcal{G}_{L'}(\Theta)}\sup_{G\in\mathcal{G}_{L'}(\Theta)\backslash\mathcal{G}_{L-1}(\Theta)}\mathbb{E}_{f_G}[\mathcal{L}_{2,r}(\overline{G}_n,G)]$$

$$\gtrsim \frac{\mathcal{L}_{2,r}(G'_*,G_*)}{8}\exp(-n\mathbb{E}_{\boldsymbol{X}\sim\mu}[\mathrm{KL}(\mathcal{N}(f_{G'_*}(\boldsymbol{X}),\sigma^2),\mathcal{N}(f_{G_*}(\boldsymbol{X}),\sigma^2))])$$

$$\gtrsim \varepsilon\cdot\exp(-n\|f_{G'_*}-f_{G_*}\|^2_{L^2(\mu)}),$$

$$\gtrsim \varepsilon\cdot\exp(-C_1 n\varepsilon^2), \tag{30}$$

where the second inequality is due to the fact that

$$\mathrm{KL}(\mathcal{N}(f_{G'_*}(\boldsymbol{X}),\sigma^2),\mathcal{N}(f_{G_*}(\boldsymbol{X}),\sigma^2)) = \frac{(f_{G'_*}(\boldsymbol{X})-f_{G_*}(\boldsymbol{X}))^2}{2\sigma^2}.$$

By choosing $\varepsilon = n^{-1/2}$, we obtain that $\varepsilon\cdot\exp(-C_1 n\varepsilon^2) = n^{-1/2}\exp(-C_1)$. As a consequence, we achieve the desired minimax lower bound in equation (29). $\qquad\square$

**Main proof.** We need to prove that the following limit holds true for any $r\geq 1$:

$$\lim_{\varepsilon\to 0}\inf_{G\in\mathcal{G}_{L'}(\Theta):\mathcal{L}_{2,r}(G,G_*)\leq\varepsilon}\frac{\|f_G-f_{G_*}\|_{L_2(\mu)}}{\mathcal{L}_{2,r}(G,G_*)} = 0. \tag{31}$$

For that purpose, it suffices to build a sequence of mixing measures $(G_n)_{n\geq 1}$ such that both $\mathcal{L}_{2,r}(G_n,G_*)\to 0$ and

$$\frac{\|f_{G_n}-f_{G_*}\|_{L_2(\mu)}}{\mathcal{L}_{2,r}(G_n,G_*)} \to 0,$$

as $n\to\infty$. To this end, we consider the sequence $G_n = \sum_{i=1}^{L+1}\exp(\beta_{0i}^n)\delta_{(\beta_{1i}^n,a_i^n,b_i^n)}$, where

- $\exp(\beta_{01}^n) = \exp(\beta_{02}^n) = \frac{1}{2}\exp(\beta_{01}^*) + \frac{1}{2n^{r+1}}$ and $\exp(\beta_{0i}^n) = \exp(\beta_{0(i-1)}^n)$ for any $3\leq i\leq L+1$;

- $\beta_{11}^n = \beta_{12}^n = \beta_{11}^*$ and $\beta_{1i}^n = \beta_{1(i-1)}^n$ for any $3\leq i\leq L+1$;

- $a_1^n = a_2^n = a_1^*$ and $a_i^n = a_{i-1}^n$ for any $3\leq i\leq L+1$;

- $b_1^n = b_1^* + \frac{1}{n}$, $b_2^n = b_1^* - \frac{1}{n}$ and $b_i^n = b_{i-1}^*$ for any $3\leq i\leq L+1$.

As a result, the loss function $\mathcal{L}_{2,r}(G_n,G_*)$ is reduced to

$$\mathcal{L}_{2,r}(G_n,G_*) = \frac{1}{n^{r+1}} + \left[\exp(\beta_{01}^*) + \frac{1}{n^{r+1}}\right]\cdot\frac{1}{n^r} = \mathcal{O}(n^{-r}). \tag{32}$$

which indicates indicates that $\mathcal{L}_{2,r}(G_n,G_*)\to 0$ as $n\to\infty$.

Now, we prove that $\|f_{G_n}-f_{G_*}\|_{L_2(\mu)}/\mathcal{L}_{2,r}(G_n,G_*)\to 0$. For that purpose, let us consider the quantity

$$Q_n(\boldsymbol{X}) := \left[\sum_{i'=1}^N\exp(\boldsymbol{X}^\top B_{i'}^0\boldsymbol{X}+c_{i'}^0) + \sum_{j'=1}^L\exp((\beta_{1j'}^*)^\top\boldsymbol{X}+\beta_{0j'}^*)\right]\cdot[g_{G_n}(\boldsymbol{X})-g_{G_*}(\boldsymbol{X})].$$

For simplicity, let us consider the polynomial degree $p=1$ as the arguments for other values of $p$ can be adapted accordingly. Recall from equation (46) that $Q_n(\boldsymbol{X})$ can be decomposed as follows:

$$Q_n(\boldsymbol{X}) = \sum_{j=1}^L\sum_{i\in\mathcal{A}_j}\exp(\beta_{0i}^n)\left[\exp((\beta_{1i}^n)^\top\boldsymbol{X})((a_i^n)^\top\boldsymbol{X}+b_i^n)-\exp((\beta_{1j}^*)^\top\boldsymbol{X})((a_j^*)^\top\boldsymbol{X}+b_j^*)\right]$$

$$-\sum_{j=1}^L\sum_{i\in\mathcal{A}_j}\exp(\beta_{0i}^n)\left[\exp((\beta_{1i}^n)^\top\boldsymbol{X})g_{G_n}(\boldsymbol{X})-\exp((\beta_{1j}^*)^\top\boldsymbol{X})g_{G_n}(\boldsymbol{X})\right]$$

$$+\sum_{j=1}^L\left(\sum_{i\in\mathcal{A}_j}\exp(\beta_{0i}^n)-\exp(\beta_{0j}^*)\right)\left[\exp((\beta_{1j}^*)^\top\boldsymbol{X})((a_j^*)^\top\boldsymbol{X}+b_j^*)-\exp((\beta_{1j}^*)^\top\boldsymbol{X})g_{G_n}(\boldsymbol{X})\right]$$

$$:= A_n(\boldsymbol{X}) - B_n(\boldsymbol{X}) + C_n(\boldsymbol{X}).$$

From the definitions of $\beta_{1i}^n$, $a_i^n$ and $b_i^n$, we can rewrite $A_n(\boldsymbol{X})$ as follows:

$$
\begin{aligned}
A_n(\boldsymbol{X}) &= \sum_{i=1}^2 \frac{1}{2}\Big[\exp(\beta_{01}^*) + \frac{1}{n^{r+1}}\Big]\exp((\beta_{11}^*)^\top \boldsymbol{X})[((a_i^n)^\top \boldsymbol{X} + b_i^n) - ((a_1^*)^\top \boldsymbol{X} + b_1^*)] \\
&= \frac{1}{2}\Big[\exp(\beta_{01}^*) + \frac{1}{n^{r+1}}\Big]\exp((\beta_{11}^*)^\top \boldsymbol{X})[(b_1^n - b_1^*) + (b_2^n - b_1^*)] \\
&= 0.
\end{aligned}
$$

Additionally, it can also be checked that $B_n(\boldsymbol{X}) = 0$, and $C_n(\boldsymbol{X}) = \mathcal{O}(n^{-(r+1)})$. Therefore, it follows that $C_n(\boldsymbol{X})/\mathcal{L}_{2,r}(G_n, G_*) \to 0$. As a consequence, $Q_n(\boldsymbol{X})/\mathcal{L}_{2,r}(G_n, G_*) \to 0$ as $n \to \infty$ for almost every $\boldsymbol{X}$.

Since the term $\Big[\sum_{i'=1}^N \exp(\boldsymbol{X}^\top B_{i'}^0 \boldsymbol{X} + c_{i'}^0) + \sum_{j'=1}^L \exp((\beta_{1j'}^*)^\top \boldsymbol{X} + \beta_{0j'}^*)\Big]$ is bounded, we deduce that $[f_{G_n}(\boldsymbol{X}) - f_{G_*}(\boldsymbol{X})]/\mathcal{L}_{2,r} \to 0$ for almost every $\boldsymbol{X}$. This result indicates that

$$
\|f_{G_n} - f_{G_*}\|_{L_2(\mu)}/\mathcal{L}_{2,r}(G_n, G_*) \to 0
$$

as $n \to \infty$. Hence, the proof of claim (31) is completed. $\qquad\square$

# B  Proof of Theoretical Results

In this appendix, we present rigorous proofs for the theoretical results introduced in Section 4, namely Theorem 4.1 and Theorem 4.3, in that order.

## B.1  Proof of Theorem 4.1

For the proof of the theorem, we first introduce some notation. Firstly, we denote by $\mathcal{F}_{L'}(\Theta)$ the set of conditional densities of all mixing measures in $\mathcal{G}_{L'}(\Theta)$, that is, $\mathcal{F}_{L'}(\Theta) := \{g_G(\boldsymbol{X}) : G \in \mathcal{G}_{L'}(\Theta)\}$. Additionally, for each $\delta > 0$, the $L^2(\mu)$ ball centered around the conditional density $g_{G_*}(Y|X)$ and intersected with the set $\mathcal{F}_{L'}(\Theta)$ is defined as

$$
\mathcal{F}_{L'}(\Theta, \delta) := \big\{g \in \mathcal{F}_{L'}(\Theta) : \|g - g_{G_*}\|_{L^2(\mu)} \leq \delta\big\}.
$$

In order to measure the size of the above set, Geer et. al. [45] suggest using the following quantity:

$$
\mathcal{J}_B(\delta, \mathcal{F}_{L'}(\Theta, \delta)) := \int_{\delta^2/2^{13}}^{\delta} H_B^{1/2}(t, \mathcal{F}_{L'}(\Theta, t), \|\cdot\|_{L^2(\mu)})\, \mathrm{d}t \vee \delta, \tag{33}
$$

where $H_B(t, \mathcal{F}_{L'}(\Theta, t), \|\cdot\|_{L^2(\mu)})$ stands for the bracketing entropy [45] of $\mathcal{F}_{L'}(\Theta, u)$ under the $L^2(\mu)$-norm, and $t \vee \delta := \max\{t, \delta\}$. By using the similar proof argument of Theorem 7.4 and Theorem 9.2 in [45] with notations being adapted to this work, we obtain the following lemma:

**Lemma B.1.** *Take $\Psi(\delta) \geq \mathcal{J}_B(\delta, \mathcal{F}_{L'}(\Theta, \delta))$ that satisfies $\Psi(\delta)/\delta^2$ is a non-increasing function of $\delta$. Then, for some universal constant $c$ and for some sequence $(\delta_n)$ such that $\sqrt{n}\delta_n^2 \geq c\Psi(\delta_n)$, we achieve that*

$$
\mathbb{P}\Big(\|g_{\widehat{G}_n} - g_{G_*}\|_{L^2(\mu)} > \delta\Big) \leq c \exp\Big(-\frac{n\delta^2}{c^2}\Big),
$$

*for all $\delta \geq \delta_n$.*

We now demonstrate that when the expert functions are Lipschitz continuous, the following bound holds:

$$
H_B(\varepsilon, \mathcal{F}_{L'}(\Theta), \|\cdot\|_{L^2(\mu)}) \lesssim \log(1/\varepsilon), \tag{34}
$$

for any $0 < \varepsilon \leq 1/2$. Indeed, for any function $g_G \in \mathcal{F}_{L'}(\Theta)$, since the expert functions are bounded, we obtain that $h(\boldsymbol{X}, \eta) \leq M$ for all $\boldsymbol{X}$ where $M$ is a bounded constant of the expert functions. Let $\tau \leq \varepsilon$ and $\{\pi_1, \ldots, \pi_{\bar{N}}\}$ be the $\zeta$-cover under the $L^\infty$ norm of the set $\mathcal{F}_{L'}(\Theta)$ where $\bar{N} := N(\zeta, \mathcal{F}_{L'}(\Theta), \|\cdot\|_{L^\infty})$ is the $\eta$-covering number of the metric space $(\mathcal{F}_{L'}(\Theta), \|\cdot\|_{L^\infty})$. Then, we construct the brackets of the form $[L_i(\boldsymbol{X}), U_i(\boldsymbol{X})]$ for all $i \in [\bar{N}]$ as follows:

$$
\begin{aligned}
L_i(x) &:= \max\{\pi_i(\boldsymbol{X}) - \zeta, 0\}, \\
U_i(x) &:= \max\{\pi_i(\boldsymbol{X}) + \zeta, M\}.
\end{aligned}
$$

From the above construction, we can validate that $\mathcal{F}_{L'}(\Theta) \subset \cup_{i=1}^{\bar{N}}[L_i(\boldsymbol{X}), U_i(\boldsymbol{X})]$ and $U_i(\boldsymbol{X}) - L_i(\boldsymbol{X}) \leq \min\{2\zeta, M\}$. Therefore, it follows that

$$\|U_i - L_i\|_{L_2(\mu)}^2 = \int (U_i - L_i)^2 \mathrm{d}\mu(\boldsymbol{X}) \leq \int 4\zeta^2 \mathrm{d}\mu(\boldsymbol{X}) = 4\zeta^2,$$

which implies that $\|U_i - L_i\|_{L_2(\mu)} \leq 2\zeta$. By definition of the bracketing entropy, we deduce that

$$H_B(2\zeta, \mathcal{F}_{L'}(\Theta), \|\cdot\|_{L_2(\mu)}) \leq \log N = \log N(\zeta, \mathcal{F}_{L'}(\Theta), \|\cdot\|_{L^\infty}). \tag{35}$$

Therefore, we need to provide an upper bound for the covering number $\bar{N}$. In particular, we denote $\Delta := \{(\beta_1, \beta_0) \in \mathbb{R}^{Nd \times Nd} \times \mathbb{R}^{Nd} \times \mathbb{R} : (\beta_1, \beta_0, \eta) \in \Theta\}$ and $\Omega := \{\eta \in \mathbb{R}^q : (\beta_1, \beta_0, \eta) \in \Theta\}$. Since $\Theta$ is a compact set, $\Delta$ and $\Omega$ are also compact. Therefore, we can find $\zeta$-covers $\Delta_\zeta$ and $\Omega_\zeta$ for $\Delta$ and $\Omega$, respectively. We can check that

$$|\Delta_\zeta| \leq \mathcal{O}_P(\tau^{-(Nd+1)L'}), \quad |\Omega_\zeta| \lesssim \mathcal{O}_P(\tau^{-qL'}).$$

For each mixing measure $G = \sum_{i=1}^{L'} \exp(\beta_{0i})\delta_{(\beta_{1i}, \eta_i)} \in \mathcal{G}_{L'}(\Theta)$, we consider other two mixing measures:

$$\check{G} := \sum_{i=1}^{L'} \exp(\beta_{0i})\delta_{(\beta_{1i}, \overline{\eta}_i)}, \qquad \overline{G} := \sum_{i=1}^{L'} \exp(\overline{\beta}_{0i})\delta_{(\overline{\beta}_{1i}, \overline{\eta}_i)}.$$

Here, $\overline{\eta}_i \in \Omega_\zeta$ such that $\overline{\eta}_i$ is the closest to $\eta_i$ in that set, while $(\overline{\beta}_{1i}, \overline{\beta}_{0i}) \in \Delta_\zeta$ is the closest to $(\beta_{1i}, \beta_{0i})$ in that set. From the above formulations, we get that

$$
\begin{aligned}
&\|g_G - g_{\check{G}}\|_{L^\infty} \\
&= \sup_{\boldsymbol{X} \in \mathcal{X}} \sum_{j=1}^{L'} \frac{\exp(\beta_{1j}^\top \boldsymbol{X} + \alpha\sigma(\tau\beta_{1j}^\top \boldsymbol{X}) + \beta_{0j}) \cdot |h(\boldsymbol{X}, \eta_j) - h(\boldsymbol{X}, \overline{\eta}_j)|}{\sum_{i'=1}^{N} \exp(\boldsymbol{X}^\top B_{i'}^0 \boldsymbol{X} + c_{i'}^0) + \sum_{j'=1}^{L'} \exp(\beta_{1j'}^\top \boldsymbol{X} + \alpha\sigma(\tau\beta_{1j'}^\top x) + \beta_{0j'})} \\
&\leq \sum_{j=1}^{L'} \sup_{\boldsymbol{X} \in \mathcal{X}} \frac{\exp(\beta_{1j}^\top \boldsymbol{X} + \alpha\sigma(\tau\beta_{1j}^\top \boldsymbol{X}) + \beta_{0j}) \cdot |h(\boldsymbol{X}, \eta_j) - h(\boldsymbol{X}, \overline{\eta}_j)|}{\sum_{i'=1}^{N} \exp(\boldsymbol{X}^\top B_{i'}^0 \boldsymbol{X} + c_{i'}^0) + \sum_{j'=1}^{L'} \exp(\beta_{1j'}^\top \boldsymbol{X} + \alpha\sigma(\tau\beta_{1j'}^\top \boldsymbol{X}) + \beta_{0j'})} \\
&\leq \sum_{j=1}^{L'} \sup_{\boldsymbol{X} \in \mathcal{X}} |h(\boldsymbol{X}, \eta_j) - h(\boldsymbol{X}, \overline{\eta}_j)| \\
&\leq \sum_{j=1}^{L'} \sup_{\boldsymbol{X} \in \mathcal{X}} [L_1(\boldsymbol{X}) \cdot \|\eta_j - \overline{\eta}_j\|] \\
&\lesssim L'\zeta \lesssim \zeta.
\end{aligned}
$$

Here, the second inequality occurs as the softmax weight is bounded by one, and the third inequality follows from the fact that the expert $h(\boldsymbol{X}, \cdot)$ is a Lipschitz function with some Lipschitz constant $L_1(\boldsymbol{X}) > 0$. Next, let us denote

$$D := \sum_{i'=1}^{N} \exp(\boldsymbol{X}^\top B_{i'}^0 \boldsymbol{X} + c_{i'}^0) + \sum_{j'=1}^{L'} \exp(\beta_{1j'}^\top \boldsymbol{X} + \alpha\sigma(\tau\beta_{1j'}^\top \boldsymbol{X}) + \beta_{0j'}),$$

$$\overline{D} := \sum_{i'=1}^{N} \exp(\boldsymbol{X}^\top B_{i'}^0 \boldsymbol{X} + c_{i'}^0) + \sum_{j'=1}^{L'} \exp(\overline{\beta}_{1j'}^\top \boldsymbol{X} + \alpha\sigma(\tau\overline{\beta}_{1j'}^\top \boldsymbol{X}) + \overline{\beta}_{0j'}).$$

Then, we have

$$\|g_{\check{G}} - g_{\overline{G}}\|_{L^\infty}$$

$$= \sup_{\boldsymbol{X} \in \mathcal{X}} \left| \frac{1}{D} \Big( \sum_{i=1}^{N} \exp(\boldsymbol{X}^\top B_i^0 \boldsymbol{X} + c_i^0) h(\boldsymbol{X}, \eta_i^0) + \sum_{j=1}^{L'} \exp(\beta_{1j}^\top \boldsymbol{X} + \alpha\sigma(\tau\beta_{1j}^\top \boldsymbol{X}) + \beta_{0j}) h(\boldsymbol{X}, \overline{\eta}_j) \Big) \right.$$

$$\left. - \frac{1}{\overline{D}} \Big( \sum_{i=1}^{N} \exp(\boldsymbol{X}^\top B_i^0 \boldsymbol{X} + c_i^0) h(\boldsymbol{X}, \eta_i^0) + \sum_{j=1}^{L'} \exp(\overline{\beta}_{1j}^\top \boldsymbol{X} + \alpha\sigma(\tau\overline{\beta}_{1j}^\top \boldsymbol{X}) + \overline{\beta}_{0j}) h(\boldsymbol{X}, \overline{\eta}_j) \Big) \right|$$

$$\leq \left| \frac{1}{D} - \frac{1}{\overline{D}} \right| \cdot \sum_{i=1}^{N} \sup_{\boldsymbol{X} \in \mathcal{X}} \left| \exp(\boldsymbol{X}^\top B_i^0 \boldsymbol{X} + c_i^0) h(\boldsymbol{X}, \eta_i^0) \right|$$

$$+ \sum_{j=1}^{L'} \sup_{\boldsymbol{X} \in \mathcal{X}} \left| \frac{\exp(\beta_{1j}^\top \boldsymbol{X} + \alpha\sigma(\tau\beta_{1j}^\top \boldsymbol{X}) + \beta_{0j})}{D} - \frac{\exp(\overline{\beta}_{1j}^\top \boldsymbol{X} + \alpha\sigma(\tau\overline{\beta}_{1j}^\top \boldsymbol{X}) + \overline{\beta}_{0j})}{\overline{D}} \right| \cdot |h(\boldsymbol{X}, \overline{\eta}_j)|.$$

$$(36)$$

Now, we will bound two terms in the above right hand side. Firstly, since both the input space $\mathcal{X}$ and the parameter space $\Theta$ are bounded, we have that

$$\frac{1}{D} - \frac{1}{\overline{D}} \lesssim |D - \overline{D}|$$

$$\leq \sum_{j'=1}^{L'} \left| \exp(\beta_{1j'}^\top \boldsymbol{X} + \alpha\sigma(\tau\beta_{1j'}^\top \boldsymbol{X}) + \beta_{0j'}) - \exp(\overline{\beta}_{1j'}^\top \boldsymbol{X} + \alpha\sigma(\tau\overline{\beta}_{1j'}^\top \boldsymbol{X}) + \overline{\beta}_{0j'}) \right|$$

$$\lesssim \sum_{j'=1}^{L'} \left| (\beta_{1j} - \overline{\beta}_{1j})^\top \boldsymbol{X} + \alpha[\sigma(\tau\beta_{1j'}^\top \boldsymbol{X}) - \sigma(\tau\overline{\beta}_{1j'}^\top \boldsymbol{X})] + (\beta_{0j} - \overline{\beta}_{0j}) \right|$$

$$\leq \sum_{j'=1}^{L'} |(\beta_{1j} - \overline{\beta}_{1j})^\top \boldsymbol{X}| + |\alpha| \cdot |\sigma(\tau\beta_{1j'}^\top \boldsymbol{X}) - \sigma(\tau\overline{\beta}_{1j'}^\top \boldsymbol{X})| + |\beta_{0j} - \overline{\beta}_{0j}|$$

$$\lesssim \sum_{j=1}^{L'} \left[ \|\beta_{1j} - \overline{\beta}_{1j}\| \cdot \|\boldsymbol{X}\| + |\alpha\tau| \cdot \|\beta_{1j} - \overline{\beta}_{1j}\| \cdot \|\boldsymbol{X}\| + |\beta_{0j} - \overline{\beta}_{0j}| \right]$$

$$\leq L'(B + |\alpha\tau|B + 1)\zeta \lesssim \zeta.$$

As a result, we deduce that

$$\left| \frac{1}{D} - \frac{1}{\overline{D}} \right| \cdot \sum_{i=1}^{N} \sup_{\boldsymbol{X} \in \mathcal{X}} \left| \exp(\boldsymbol{X}^\top B_i^0 \boldsymbol{X} + c_i^0) h(\boldsymbol{X}, \eta_i^0) \right| \lesssim \zeta. \qquad (37)$$

Regarding the second term, note that

$$\frac{\exp(\beta_{1j}^\top \boldsymbol{X} + \alpha\sigma(\tau\beta_{1j}^\top \boldsymbol{X}) + \beta_{0j})}{D} - \frac{\exp(\overline{\beta}_{1j}^\top \boldsymbol{X} + \alpha\sigma(\tau\overline{\beta}_{1j}^\top \boldsymbol{X}) + \overline{\beta}_{0j})}{\overline{D}}$$

$$= \exp(\beta_{1j}^\top \boldsymbol{X} + \alpha\sigma(\tau\beta_{1j}^\top \boldsymbol{X}) + \beta_{0j}) \Big( \frac{1}{D} - \frac{1}{\overline{D}} \Big)$$

$$+ \frac{1}{\overline{D}} \Big[ \exp(\beta_{1j}^\top \boldsymbol{X} + \alpha\sigma(\tau\beta_{1j}^\top \boldsymbol{X}) + \beta_{0j}) - \exp(\exp(\overline{\beta}_{1j}^\top \boldsymbol{X} + \alpha\sigma(\tau\overline{\beta}_{1j}^\top \boldsymbol{X}) + \overline{\beta}_{0j})) \Big].$$

Since both the input space and the parameter space are bounded, we have

$$\exp(\beta_{1j}^\top \boldsymbol{X} + \alpha\sigma(\tau\beta_{1j}^\top \boldsymbol{X}) + \beta_{0j}) \Big( \frac{1}{D} - \frac{1}{\overline{D}} \Big) \lesssim \frac{1}{D} - \frac{1}{\overline{D}} \lesssim \zeta,$$

$$\frac{1}{\overline{D}} \Big[ \exp(\beta_{1j}^\top \boldsymbol{X} + \alpha\sigma(\tau\beta_{1j}^\top \boldsymbol{X}) + \beta_{0j}) - \exp(\overline{\beta}_{1j}^\top \boldsymbol{X} + \alpha\sigma(\tau\overline{\beta}_{1j}^\top \boldsymbol{X}) + \overline{\beta}_{0j})$$

$$\lesssim (B + |\alpha\tau|B + 1)\zeta \lesssim \zeta,$$

which yields that

$$\sum_{j=1}^{L'} \sup_{\boldsymbol{X} \in \mathcal{X}} \left| \frac{\exp(\beta_{1j}^\top \boldsymbol{X} + \alpha\sigma(\tau\beta_{1j}^\top \boldsymbol{X}) + \beta_{0j})}{D} - \frac{\exp(\overline{\beta}_{1j}^\top \boldsymbol{X} + \alpha\sigma(\tau\overline{\beta}_{1j}^\top \boldsymbol{X}) + \overline{\beta}_{0j})}{\overline{D}} \right| \cdot |h(\boldsymbol{X}, \overline{\eta}_j)|$$

$$\lesssim \zeta \sum_{j=1}^{L'} \sup_{\boldsymbol{X} \in \mathcal{X}} |h(\boldsymbol{X}, \overline{\eta}_j)| \lesssim \zeta. \tag{38}$$

From equations (36), (37) and (38), we obtain that $\|g_{\check{G}} - g_{\overline{G}}\|_{L^\infty} \lesssim \zeta$. According to the triangle inequality, we have

$$\|g_G - g_{\overline{G}}\|_{L^\infty} \leq \|g_G - g_{\check{G}}\|_{L^\infty} + \|g_{\check{G}} - g_{\overline{G}}\|_{L^\infty} \lesssim \zeta.$$

By definition of the covering number, we deduce that

$$N(\zeta, \mathcal{F}_{L'}(\Theta), \|\cdot\|_{L^\infty}) \leq |\Delta_\zeta| \times |\Omega_\zeta| \leq \mathcal{O}(n^{-(Nd+1)L'}) \times \mathcal{O}(n^{-qL'}) \leq \mathcal{O}(n^{-(Nd+1+q)L'}). \tag{39}$$

Combine equations (35) and (39), we achieve that

$$H_B(2\zeta, \mathcal{F}_{L'}(\Theta), \|\cdot\|_{L_2(\mu)}) \lesssim \log(1/\tau).$$

Let $\zeta = \varepsilon/2$, then we obtain that

$$H_B(\varepsilon, \mathcal{F}_{L'}(\Theta), \|\cdot\|_{L^2(\mu)}) \lesssim \log(1/\varepsilon).$$

As a result, it follows that

$$\mathcal{J}_B(\delta, \mathcal{F}_{L'}(\Theta, \delta)) = \int_{\delta^2/2^{13}}^\delta H_B^{1/2}(t, \mathcal{F}_{L'}(\Theta, t), \|\cdot\|_{L_2(\mu)}) \, \mathrm{d}t \vee \delta \lesssim \int_{\delta^2/2^{13}}^\delta \log(1/t) dt \vee \delta. \tag{40}$$

Let $\Psi(\delta) = \delta \cdot [\log(1/\delta)]^{1/2}$, then $\Psi(\delta)/\delta^2$ is a non-increasing function of $\delta$. Furthermore, equation (40) indicates that $\Psi(\delta) \geq \mathcal{J}_B(\delta, \mathcal{F}_{L'}(\Theta, \delta))$. In addition, let $\delta_n = \sqrt{\log(n)/n}$, then we get that $\sqrt{n}\delta_n^2 \geq c\Psi(\delta_n)$ for some universal constant $c$. Finally, by applying Lemma B.1, we achieve the desired conclusion of the theorem.

## B.2 Proof of Theorem 4.3

Our goal is also to demonstrate the following inequality:

$$\inf_{G \in \mathcal{G}_{L'}(\Theta)} \|g_G - g_{G_*}\|_{L_2(\mu)}/\mathcal{L}_1(G, G_*) > 0. \tag{41}$$

For that purpose, we divide the proof of the above inequality into local and global parts in the sequel.

**Local part:** In this part, we demonstrate that

$$\lim_{\varepsilon \to 0} \inf_{G \in \mathcal{G}_{L'}(\Theta):\mathcal{L}_1(G,G_*) \leq \varepsilon} \|g_G - g_{G_*}\|_{L_2(\mu)}/\mathcal{L}_1(G, G_*) > 0. \tag{42}$$

Assume by contrary that the above claim is not true, then there exists a sequence of mixing measures $G_n = \sum_{i=1}^{L} \exp(\beta_{0i}^n)\delta_{(\beta_{1i}^n, \eta_i^n)}$ in $\mathcal{G}_{L'}(\Theta)$ such that $\mathcal{L}_{1n} := \mathcal{L}_1(G_n, G_*) \to 0$ and

$$\|g_{G_n} - g_{G_*}\|_{L_2(\mu)}/\mathcal{L}_{1n} \to 0, \tag{43}$$

as $n \to \infty$. Let us denote by $\mathcal{V}_j^n := \mathcal{V}_j(G_n)$ a Voronoi cell of $G_n$ generated by the $j$-th components of $G_*$. Since our arguments are asymptotic, we may assume that those Voronoi cells do not depend on the sample size, i.e., $\mathcal{V}_j = \mathcal{V}_j^n$. Thus, the Voronoi loss $\mathcal{L}_{1n}$ can be represented as

$$\mathcal{L}_{1n} := \sum_{j:|\mathcal{V}_j|>1} \sum_{i \in \mathcal{V}_j} \exp(\beta_{0i}^n)\left[\|\Delta\beta_{1ij}^n\|^2 + \|\Delta\eta_{ij}^n\|^2\right]$$

$$+ \sum_{j:|\mathcal{V}_j|=1} \sum_{i \in \mathcal{V}_j} \exp(\beta_{0i}^n)\left[\|\Delta\beta_{1ij}^n\| + \|\Delta\eta_{ij}^n\|\right] + \sum_{j=1}^{k_*} \left| \sum_{i \in \mathcal{V}_j} \exp(\beta_{1i}^n) - \exp(\beta_{1j}^*) \right|, \tag{44}$$

where we denote $\Delta\beta_{1ij}^n := \beta_{1i}^n - \beta_{1j}^*$ and $\Delta\eta_{ij}^n := \eta_i^n - \eta_j^*$.

Since $\mathcal{L}_{1n} \to 0$, we get that $(\beta_{1i}^n, \eta_i^n) \to (\beta_{1j}^*, \eta_j^*)$ and $\sum_{i\in\mathcal{V}_j} \exp(\beta_{0i}^n) \to \exp(\beta_{0j}^*)$ as $n \to \infty$ for any $i \in \mathcal{V}_j$ and $j \in [L]$. Now, we divide the proof of the local part into three steps as follows:

**Step 1 - Taylor expansion.** In this step, we would like to decompose the quantity

$$Q_n(\boldsymbol{X}) := \Big[ \sum_{i'=1}^N \exp(\boldsymbol{X}^\top A_{i'}^0 \boldsymbol{X} + c_{i'}^0) + \sum_{j'=1}^L \exp((\beta_{1j'}^*)^\top \boldsymbol{X} + \alpha\sigma(\tau(\beta_{1j'}^*)^\top \boldsymbol{X}) + \beta_{0j'}^*) \Big]$$

$$\times [g_{G_n}(\boldsymbol{X}) - g_{G_*}(\boldsymbol{X})] \quad (45)$$

into a combination of linearly independent elements using Taylor expansion. In particular, the quantity $Q_n(\boldsymbol{X})$ is decomposed as follows:

$$\sum_{j=1}^L \sum_{i\in\mathcal{V}_j} \exp(\beta_{0i}^n)\Big[ \exp((\beta_{1i}^n)^\top \boldsymbol{X} + \alpha\sigma(\tau(\beta_{1i}^n)^\top \boldsymbol{X}))h(\boldsymbol{X};\eta_i^n) - \exp((\beta_{1j}^*)^\top \boldsymbol{X} + \alpha\sigma(\tau(\beta_{1j}^*)^\top \boldsymbol{X}))h(\boldsymbol{X};\eta_j^*) \Big]$$

$$-\sum_{j=1}^L \sum_{i\in\mathcal{V}_j} \exp(\beta_{0i}^n)\Big[ \exp((\beta_{1i}^n)^\top \boldsymbol{X} + \alpha\sigma(\tau(\beta_{1i}^n)^\top \boldsymbol{X})) - \exp((\beta_{1j}^*)^\top \boldsymbol{X} + \alpha\sigma(\tau(\beta_{1j}^*)^\top \boldsymbol{X})) \Big]g_{G_n}(\boldsymbol{X})$$

$$+\sum_{j=1}^L \Big( \sum_{i\in\mathcal{V}_j} \exp(\beta_{0i}^n) - \exp(\beta_{0j}^*) \Big) \exp((\beta_{1j}^*)^\top \boldsymbol{X} + \alpha\sigma(\tau(\beta_{1j}^*)^\top \boldsymbol{X}))\Big[ h(\boldsymbol{X};\eta_j^*) - g_{G_n}(\boldsymbol{X}) \Big]$$

$$:= A_n(\boldsymbol{X}) - B_n(\boldsymbol{X}) + C_n(\boldsymbol{X}). \quad (46)$$

**Decomposition of $A_n(\boldsymbol{X})$.** Let us denote $E(\boldsymbol{X};\beta_1) := \exp(\beta_1^\top \boldsymbol{X} + \alpha\sigma(\tau\beta_1^\top \boldsymbol{X}))$, then $A_n$ can be separated into two terms as follows:

$$A_n(\boldsymbol{X}) := \sum_{j:|\mathcal{V}_j|=1} \sum_{i\in\mathcal{V}_j} \exp(\beta_{0i}^n)\Big[ E(\boldsymbol{X};\beta_{1i}^n)h(x;\eta_i^n) - E(\boldsymbol{X};\beta_{1j}^*)h(\boldsymbol{X};\eta_j^*) \Big]$$

$$+ \sum_{j:|\mathcal{V}_j|>1} \sum_{i\in\mathcal{V}_j} \exp(\beta_{0i}^n)\Big[ E(\boldsymbol{X};\beta_{1i}^n)h(\boldsymbol{X};\eta_i^n) - E(\boldsymbol{X};\beta_{1j}^*)h(\boldsymbol{X};\eta_j^*) \Big]$$

$$:= A_{n,1}(\boldsymbol{X}) + A_{n,2}(\boldsymbol{X}).$$

By means of the first-order Taylor expansion, we have

$$A_{n,1}(\boldsymbol{X}) = \sum_{j:|\mathcal{V}_j|=1} \sum_{i\in\mathcal{V}_j} \frac{\exp(\beta_{0i}^n)}{\alpha!} \sum_{|\alpha|=1} (\Delta\beta_{1ij}^n)^{\alpha_1}(\Delta\eta_{ij}^n)^{\alpha_2} \frac{\partial^{|\alpha_1|}E}{\partial\beta_1^{\alpha_1}}(\boldsymbol{X};\beta_{1j}^*)\frac{\partial^{|\alpha_2|}h}{\partial\eta^{\alpha_2}}(\boldsymbol{X};\eta_j^*) + R_{n,1}(\boldsymbol{X})$$

$$= \sum_{j:|\mathcal{V}_j|=1} \sum_{|\alpha_1|+|\alpha_2|=1} S_{n,j,\alpha_1,\alpha_2} \frac{\partial^{|\alpha_1|}E}{\partial\beta_1^{\alpha_1}}(\boldsymbol{X};\beta_{1j}^*)\frac{\partial^{|\alpha_2|}h}{\partial\eta^{\alpha_2}}(\boldsymbol{X};\eta_j^*) + R_{n,1}(\boldsymbol{X}),$$

where $R_{n,1}(\boldsymbol{X})$ is a Taylor remainder such that $R_{n,1}(\boldsymbol{X})/\mathcal{L}_{1n} \to 0$ as $n \to \infty$, and

$$S_{n,j,\alpha_1,\alpha_2} := \sum_{i\in\mathcal{V}_j} \frac{\exp(\beta_{0i}^n)}{\alpha!}(\Delta\beta_{1ij}^n)^{\alpha_1}(\Delta\eta_{ij}^n)^{\alpha_2}.$$

On the other hand, by applying the second-order Taylor expansion, we get that

$$A_{n,2}(\boldsymbol{X}) = \sum_{j:|\mathcal{V}_j|>1} \sum_{1\le|\alpha_1|+|\alpha_2|\le 2} S_{n,j,\alpha_1,\alpha_2} \frac{\partial^{|\alpha_1|}E}{\partial\beta_1^{\alpha_1}}(\boldsymbol{X};\beta_{1j}^*)\frac{\partial^{|\alpha_2|}h}{\partial\eta^{\alpha_2}}(\boldsymbol{X};\eta_j^*) + R_{n,2}(\boldsymbol{X}),$$

in which $R_{n,2}(\boldsymbol{X})$ is a Taylor remainder such that $R_{n,2}(\boldsymbol{X})/\mathcal{L}_{1n} \to 0$ as $n \to \infty$.

**Decomposition of $B_n$.** Recall that we have

$$B_n(\boldsymbol{X}) = \sum_{j:|\mathcal{V}_j|=1} \sum_{i\in\mathcal{V}_j} \exp(\beta_{0i}^n)\Big[ E(\boldsymbol{X};\beta_{1i}^n) - E(\boldsymbol{X};\beta_{1j}^*) \Big]g_{G_n}(\boldsymbol{X})$$

$$+ \sum_{j:|\mathcal{V}_j|>1} \sum_{i\in\mathcal{V}_j} \exp(\beta_{0i}^n)\Big[ E(\boldsymbol{X};\beta_{1i}^n) - E(x;\beta_{1j}^*) \Big]g_{G_n}(\boldsymbol{X})$$

$$:= B_{n,1}(\boldsymbol{X}) + B_{n,2}(\boldsymbol{X}).$$

By invoking first-order and second-order Taylor expansions to $B_{n,1}(\boldsymbol{X})$ and $B_{n,2}(\boldsymbol{X})$, it follows that

$$B_{n,1}(\boldsymbol{X}) = \sum_{j:|\mathcal{V}_j|=1} \sum_{|\ell|=1} T_{n,j,\ell} \cdot \frac{\partial^{|\ell|} E}{\partial \beta_1^{\ell}}(\boldsymbol{X}; \beta_{1j}^*) g_{G_n}(\boldsymbol{X}) + R_{n,3}(\boldsymbol{X}),$$

$$B_{n,2}(\boldsymbol{X}) = \sum_{j:|\mathcal{V}_j|>1} \sum_{1\leq|\ell|\leq2} T_{n,j,\ell} \cdot \frac{\partial^{|\ell|} E}{\partial \beta_1^{\ell}}(\boldsymbol{X}; \beta_{1j}^*) g_{G_n}(\boldsymbol{X}) + R_{n,4}(\boldsymbol{X}),$$

where we define

$$T_{n,j,\ell} := \sum_{i\in\mathcal{V}_j} \frac{\exp(\beta_{0i}^n)}{\ell!}(\Delta\beta_{1ij}^n)^{\ell}.$$

Additionally, $R_{n,3}(\boldsymbol{X})$ and $R_{n,4}(\boldsymbol{X})$ are Taylor remainders such that $R_{n,3}(\boldsymbol{X})/\mathcal{L}_{1n} \to 0$ and $R_{n,3}(\boldsymbol{X})/\mathcal{L}_{1n} \to 0$ as $n \to \infty$.

Collect the above results together, we can represent $Q_n(x)$ as

$$Q_n(\boldsymbol{X}) = \sum_{j=1}^{L} \sum_{0\leq|\alpha_1|+|\alpha_2|\leq2} S_{n,j,\alpha_1,\alpha_2} \frac{\partial^{|\alpha_1|} E}{\partial \beta_1^{\alpha_1}}(\boldsymbol{X}; \beta_{1j}^*) \frac{\partial^{|\alpha_2|} h}{\partial \eta^{\alpha_2}}(\boldsymbol{X}; \eta_j^*),$$

$$- \sum_{j=1}^{L} \sum_{0\leq|\ell|\leq2} T_{n,j,\ell} \cdot \frac{\partial^{|\ell|} E}{\partial \beta_1^{\ell}}(\boldsymbol{X}; \beta_{1j}^*) g_{G_n}(\boldsymbol{X}) + \sum_{i=1}^{4} R_{n,i}(\boldsymbol{X}), \qquad (47)$$

where we define $S_{n,j,\mathbf{0}_{d\times d},\mathbf{0}_q} = T_{n,j,\mathbf{0}_{d\times d}} = \sum_{i\in\mathcal{V}_j} \exp(\beta_{0i}^n) - \exp(\beta_{0j}^*)$ for any $j \in [L]$.

**Step 2 - Non-vanishing coefficients.** In this step, we demonstrate that at least one among ratios of the forms $S_{n,j,\alpha_1,\alpha_2}/\mathcal{L}_{1n}$ and $T_{n,j,\ell}/\mathcal{L}_{1n}$ goes to zero as $n$ tends to infinity. Indeed, assume by contrary that

$$\frac{S_{n,j,\alpha_1,\alpha_2}}{\mathcal{L}_{1n}} \to 0, \qquad \frac{T_{n,j,\ell}}{\mathcal{L}_{1n}} \to 0,$$

for any $j \in [L]$, $0 \leq |\alpha_1|, |\alpha_2|, |\ell| \leq 2$. Then, we get

$$\frac{1}{\mathcal{L}_{1n}} \sum_{j=1}^{L} \Big| \sum_{i\in\mathcal{V}_j} \exp(\beta_{0i}^n) - \exp(\beta_{0j}^*) \Big| = \sum_{j=1}^{L} \Big| \frac{S_{n,j,\mathbf{0}_{d\times d},\mathbf{0}_q}}{\mathcal{L}_{1n}} \Big| \to 0. \qquad (48)$$

Now, we consider indices $j \in [L]$ such that its corresponding Voronoi cell has only one element, i.e. $|\mathcal{V}_j| = 1$.

- For arbitrary $u, v \in [Nd]$, let $\alpha_1 \in \mathbb{N}^{Nd\times Nd}$ and $\alpha_2 = \mathbf{0}_q$ such that $\alpha_1^{(uv)} = 1$ while other entries equal to zero. Then, we have $\frac{1}{\mathcal{L}_{1n}} \cdot \sum_{i\in\mathcal{V}_j} \exp(\beta_{0i}^n)|(\Delta\beta_{1ij}^n)^{(uv)}| = |S_{n,j,\alpha_1,\alpha_2}|/\mathcal{L}_{1n} \to 0$ as $n \to \infty$. By taking the summation of the previous term with $u, v \in [Nd]$, we achieve that $\frac{1}{\mathcal{L}_{1n}} \sum_{i\in\mathcal{V}_j} \exp(\beta_{0i}^n)\|\Delta\beta_{1ij}^n\|_1 \to 0$. Owing to the topological equivalence between norm-1 and norm-2, it follows that

$$\frac{1}{\mathcal{L}_{1n}} \sum_{i\in\mathcal{V}_j} \exp(\beta_{0i}^n)\|\Delta\beta_{1ij}^n\| \to 0. \qquad (49)$$

- For arbitrary $u \in [Nd]$, let $\alpha_1 = \mathbf{0}_{Nd\times Nd}$ and $\alpha_2 \in \mathbb{N}^q$ such that $\alpha_2^{(u)} = 1$ while other entries equal to zero. Then, we get $\frac{1}{\mathcal{L}_{1n}} \cdot \sum_{i\in\mathcal{V}_j} \exp(\beta_{0i}^n)|(\Delta\eta_{ij}^n)^{(u)}| = |S_{n,j,\alpha_1,\alpha_2}|/\mathcal{L}_{1n} \to 0$ as $n \to \infty$. By taking the summation of the previous term with $u \in [q]$, we achieve that $\frac{1}{\mathcal{L}_{1n}} \sum_{i\in\mathcal{V}_j} \exp(\beta_{0i}^n)\|\Delta\eta_{ij}^n\|_1 \to 0$, or equivalently,

$$\frac{1}{\mathcal{L}_{1n}} \sum_{i\in\mathcal{V}_j} \exp(\beta_{0i}^n)\|\Delta\eta_{ij}^n\| \to 0. \qquad (50)$$

Combine the limits in equations (49) and (50), we obtain that

$$\frac{1}{\mathcal{L}_{1n}} \sum_{j:|\mathcal{V}_j|=1} \sum_{i \in \mathcal{V}_j} \exp(\beta_{0i}^n)[\|\Delta\beta_{1ij}^n\| + \|\Delta\eta_{ij}^n\|] \to 0, \tag{51}$$

as $n \to \infty$.

Next, we consider indices $j \in [L]$ such that its corresponding Voronoi cell has more than one element, i.e. $|\mathcal{V}_j| > 1$.

- For arbitrary $u, v \in [Nd]$, let $\alpha_1 \in \mathbb{N}^{Nd \times Nd}$ and $\alpha_2 = \mathbf{0}_q$ such that $\alpha_1^{(uv)} = 2$ while other entries equal to zero. Then, we have $\frac{1}{\mathcal{L}_{1n}} \cdot \sum_{i \in \mathcal{V}_j} \exp(\beta_{0i}^n)|(\Delta\beta_{1ij}^n)^{(uv)}|^2 = |S_{n,j,\alpha_1,\alpha_2}|/\mathcal{L}_{1n} \to 0$ as $n \to \infty$. By taking the summation of the previous term with $u, v \in [Nd]$, we achieve that

$$\frac{1}{\mathcal{L}_{1n}} \sum_{i \in \mathcal{V}_j} \exp(\beta_{0i}^n)\|\Delta\beta_{1ij}^n\|^2 \to 0. \tag{52}$$

- For arbitrary $u \in [Nd]$, let $\alpha_1 = \mathbf{0}_{Nd \times Nd}$ and $\alpha_2 \in \mathbb{N}^q$ such that $\alpha_2^{(u)} = 2$ while other entries equal to zero. Then, we get $\frac{1}{\mathcal{L}_{1n}} \cdot \sum_{i \in \mathcal{V}_j} \exp(\beta_{0i}^n)|(\Delta\eta_{ij}^n)^{(u)}|^2 = |S_{n,j,\alpha_1,\alpha_2}|/\mathcal{L}_{1n} \to 0$ as $n \to \infty$. By taking the summation of the previous term with $u \in [q]$, we achieve that

$$\frac{1}{\mathcal{L}_{1n}} \sum_{i \in \mathcal{V}_j} \exp(\beta_{0i}^n)\|\Delta\eta_{ij}^n\|^2 \to 0. \tag{53}$$

Putting the limits in equations (49) and (50), we have

$$\frac{1}{\mathcal{L}_{1n}} \sum_{j:|\mathcal{V}_j|>1} \sum_{i \in \mathcal{V}_j} \exp(\beta_{0i}^n)[\|\Delta\beta_{1ij}^n\| + \|\Delta\eta_{ij}^n\|] \to 0, \tag{54}$$

as $n \to \infty$. Taking the summation of three limits in equations (48), (51) and (54), we deduce that $1 = \mathcal{L}_{1n}/\mathcal{L}_{1n} \to 0$ as $n \to \infty$, which is a contradiction. Thus, at least one among ratios of the forms $S_{n,j,\alpha_1,\alpha_2}/\mathcal{L}_{1n}$ and $T_{n,j,\ell}/\mathcal{L}_{1n}$ goes to zero as $n$ tends to infinity.

**Step 3 - Application of Fatou's lemma.** In this step, we show that all the ratios $S_{n,j,\alpha_1,\alpha_2}/\mathcal{L}_{1n}$ and $T_{n,j,\ell}/\mathcal{L}_{1n}$ go to zero as $n \to \infty$, which contradicts to the conclusion in Step 2. In particular, by denoting $m_n$ as the maximum of the absolute values of those ratios. From the result of Step 2, it follows that $1/m_n \not\to \infty$.

Recall from the hypothesis in equation (43) that $\|g_{G_n} - g_{G_*}\|_{L_2(\mu)}/\mathcal{L}_{1n} \to 0$ as $n \to \infty$, which indicates that $\|g_{G_n} - g_{G_*}\|_{L_1(\mu)}/\mathcal{L}_{1n} \to 0$. Therefore, by applying the Fatou's lemma, we get that

$$0 = \lim_{n \to \infty} \frac{\|g_{G_n} - g_{G_*}\|_{L_1(\mu)}}{m_n \mathcal{L}_{1n}} \geq \int \liminf_{n \to \infty} \frac{|g_{G_n}(\boldsymbol{X}) - g_{G_*}(\boldsymbol{X})|}{m_n \mathcal{L}_{1n}} d\mu(\boldsymbol{X}) \geq 0.$$

This result implies that $\frac{1}{m_n \mathcal{L}_{1n}} \cdot [g_{G_n}(\boldsymbol{X}) - g_{G_*}(\boldsymbol{X})] \to 0$ as $n \to \infty$ for $\mu$-almost surely $\boldsymbol{X}$. Looking at the formulation of $Q_n(\boldsymbol{X})$ in equation (45), since the term $\left[ \sum_{i'=1}^{k_0} \exp(\boldsymbol{X}^\top A_{i'}^0 \boldsymbol{X} + c_{i'}^0) + \sum_{j'=1}^{k_*} \exp((\beta_{1j'}^*)^\top \boldsymbol{X} + \sigma((\beta_{1j'}^*)^\top \boldsymbol{X}) + \beta_{0j'}^*) \right]$ is bounded, we deduce that the term $\frac{Q_n(\boldsymbol{X})}{m_n \mathcal{L}_{1n}} \to 0$ for $\mu$-almost surely $\boldsymbol{X}$.

Let us denote

$$\frac{S_{n,j,\alpha_1,\alpha_2}}{m_n \mathcal{L}_{1n}} \to \phi_{j,\alpha_1,\alpha_2}, \qquad \frac{T_{n,j,\ell}}{m_n \mathcal{L}_{1n}} \to \varphi_{j,\ell},$$

with a note that at least one among them is non-zero. Then, from the decomposition of $Q_n(\boldsymbol{X})$ in equation (47), we have

$$\sum_{j=1}^{L} \sum_{|\alpha_1|+|\alpha_2|=0}^{1+\mathbf{1}_{\{|\mathcal{V}_j|>1\}}} \phi_{j,\alpha_1,\alpha_2} \cdot \frac{\partial^{|\alpha_1|} E}{\partial\beta_1^{\alpha_1}}(\boldsymbol{X};\beta_{1j}^*) \frac{\partial^{|\alpha_2|} h}{\partial\eta^{\alpha_2}}(\boldsymbol{X};\eta_j^*),$$

$$-\sum_{j=1}^{L} \sum_{|\ell|=0}^{1+\mathbf{1}_{\{|\mathcal{V}_j|>1\}}} \varphi_{j,\ell} \cdot \frac{\partial^{|\ell|} E}{\partial\beta_1^\ell}(\boldsymbol{X};\beta_{1j}^*) g_{G_*}(\boldsymbol{X}) = 0,$$

for $\mu$-almost surely $\boldsymbol{X}$. It is worth noting that the term $\frac{\partial^{|\alpha_1|}E}{\partial\beta_1^{\alpha_1}}(\boldsymbol{X};\beta_{1j}^*)\cdot\frac{\partial^{|\alpha_2|}h}{\partial\eta^{\alpha_2}}(\boldsymbol{X};\eta_j^*)$ can be explicitly expressed as

- When $|\alpha_1|=0, |\alpha_2|=0$: $\exp((\beta_{1j}^*)^\top\boldsymbol{X}+\sigma((\beta_{1j}^*)^\top\boldsymbol{X}))h(\boldsymbol{X};\eta_j^*)$;

- When $|\alpha_1| = 1, |\alpha_2| = 0$: $\boldsymbol{X}^{(u)}\left(1+\sigma'((\beta_{1j}^*)^\top\boldsymbol{X})\right)\exp((\beta_{1j}^*)^\top\boldsymbol{X}+\sigma((\beta_{1j}^*)^\top\boldsymbol{X}))h(\boldsymbol{X};\eta_j^*)$;

- When $|\alpha_1|=0, |\alpha_2|=1$: $\exp((\beta_{1j}^*)^\top\boldsymbol{X}+\sigma((\beta_{1j}^*)^\top\boldsymbol{X}))\frac{\partial h}{\partial\eta^{(w)}}(\boldsymbol{X};\eta_j^*)$;

- When $|\alpha_1|=1, |\alpha_2|=1$:

$$x^{(u)}\left(1+\sigma'((\beta_{1j}^*)^\top x)\right)\exp((\beta_{1j}^*)^\top x+\sigma((\beta_{1j}^*)^\top x))\frac{\partial h}{\partial\eta^{(w)}}(x;\eta_j^*);$$

- When $|\alpha_1|=2, |\alpha_2|=0$:

$$\boldsymbol{X}^{(u)}x^{(v)}\left[(1+\sigma'((\beta_{1j}^*)^\top\boldsymbol{X}))^2+\sigma''((\beta_{1j}^*)^\top\boldsymbol{X})\right]\exp((\beta_{1j}^*)^\top\boldsymbol{X}+\sigma((\beta_{1j}^*)^\top\boldsymbol{X}))h(\boldsymbol{X};\eta_j^*)$$

- When $|\alpha_1|=0, |\alpha_2|=2$: $\exp((\beta_{1j}^*)^\top\boldsymbol{X}+\sigma((\beta_{1j}^*)^\top\boldsymbol{X}))\frac{\partial^2 h}{\partial\eta^{(w)}\partial\eta^{(w')}}(\boldsymbol{X};\eta_j^*)$.

Recall that the expert function $h$ satisfies the condition in Definition 4.2, i.e. the set

$$\left\{\boldsymbol{X}^\nu\left[(1+\sigma'((\beta_{1j}^*)^\top\boldsymbol{X}))^{|\nu|}+\mathbf{1}_{\{|\nu|=2\}}\sigma''((\beta_{1j}^*)^\top\boldsymbol{X})\right]\cdot\frac{\partial^{|\gamma|}h}{\partial\eta^\gamma}(\boldsymbol{X},\eta_j^*):j\in[L],\ 0\leq|\nu|+|\gamma|\leq 2\right\}$$

is linearly independent for almost every $\boldsymbol{X}$. Therefore, we obtain that $\phi_{j,\alpha_1,\alpha_2}=\varphi_{j,\ell}=0$ for all $j\in[L], 0\leq|\alpha_1|+|\alpha_2|, |\ell|\leq 1+\mathbf{1}_{\{|\nu_j|>1\}}$. This result turns out to contradict the fact that at least one among them is different from zero. Hence, we achieve the inequality in equation (42).

**Global part.** It is worth noting that the inequality (42) suggests that there exists a positive constant $\varepsilon'$ such that

$$\inf_{G\in\mathcal{G}_{L'}(\Theta):\mathcal{L}_1(G,G_*)\leq\varepsilon'}\|g_G-g_{G_*}\|_{L_2(\mu)}/\mathcal{L}_1(G,G_*)>0.$$

Therefore, it is sufficient to prove that

$$\inf_{G\in\mathcal{G}_{L'}(\Theta):\mathcal{L}_1(G,G_*)>\varepsilon'}\|g_G-g_{G_*}\|_{L_2(\mu)}/\mathcal{L}_1(G,G_*)>0. \tag{55}$$

Assume by contrary that the inequality (55) does not hold true, then we can find a sequence of mixing measures $G'_n\in\mathcal{G}_{L'}(\Theta)$ such that $\mathcal{L}_1(G'_n,G_*)>\varepsilon'$ and

$$\lim_{n\to\infty}\frac{\|g_{G'_n}-g_{G_*}\|_{L_2(\mu)}}{\mathcal{L}_1(G'_n,G_*)}=0,$$

which indicates that $\|g_{G'_n}-g_{G_*}\|_{L_2(\mu)}\to 0$ as $n\to\infty$. Recall that $\Theta$ is a compact set, therefore, we can replace the sequence $G'_n$ by one of its subsequences that converge to a mixing measure $G'\in\mathcal{G}_{L'}(\Omega)$. Since $\mathcal{L}_1(G'_n,G_*)>\varepsilon'$, we deduce that $\mathcal{L}_1(G',G_*)>\varepsilon'$.

Next, by invoking the Fatou's lemma, we have that

$$0=\lim_{n\to\infty}\|g_{G'_n}-g_{G_*}\|_{L_2(\mu)}^2\geq\int\liminf_{n\to\infty}\left|g_{G'_n}(\boldsymbol{X})-g_{G_*}(\boldsymbol{X})\right|^2\mathrm{d}\mu(\boldsymbol{X}).$$

Thus, we get that $g_{G'}(\boldsymbol{X})=g_{G_*}(\boldsymbol{X})$ for $\mu$-almost surely $\boldsymbol{X}$. From the identifiability property of the non-linear residual gating prefix MoE (cf. the end of this proof), we deduce that $G'\equiv G_*$. Consequently, it follows that $\mathcal{L}_1(G',G_*)=0$, contradicting the fact that $\mathcal{L}_1(G',G_*)>\varepsilon'>0$. Hence, the proof is completed.

**Identifiability of Non-linear Residual Gating MoE.**

We now prove the identifiability of the non-linear residual gating prefix MoE. In particular, we will show that if $g_G(\boldsymbol{X})=g_{G_*}(\boldsymbol{X})$ for almost every $\boldsymbol{X}$, then it follows that $G\equiv G_*$.

For ease of presentation, let us denote

$$\text{softmax}_G(u) := \frac{\exp(u)}{\sum_{i'=1}^{N} \exp(\boldsymbol{X}^\top B_{i'}^0 \boldsymbol{X} + c_{i'}^0) + \sum_{j'=1}^{L} \exp((\beta_{1j'})^\top \boldsymbol{X} + \alpha\sigma(\tau(\beta_{1j'})^\top \boldsymbol{X}) + \beta_{0j'})},$$

$$\text{softmax}_{G_*}(u^*) := \frac{\exp(u^*)}{\sum_{i'=1}^{N} \exp(\boldsymbol{X}^\top B_{i'}^0 \boldsymbol{X} + c_{i'}^0) + \sum_{j'=1}^{L} \exp((\beta_{1j'}^*)^\top \boldsymbol{X} + \alpha\sigma(\tau(\beta_{1j'}^*)^\top \boldsymbol{X}) + \beta_{0j'}^*)},$$

where

$$u \in \left\{ \boldsymbol{X}^\top B_i^0 \boldsymbol{X} + c_i^0, \ (\beta_{1j'})^\top \boldsymbol{X} + \alpha\sigma(\tau(\beta_{1j'})^\top \boldsymbol{X}) + \beta_{0j'} : i' \in [N], j' \in [L'] \right\},$$

$$u^* \in \left\{ \boldsymbol{X}^\top B_i^0 \boldsymbol{X} + c_i^0, \ (\beta_{1j'}^*)^\top \boldsymbol{X} + \alpha\sigma(\tau(\beta_{1j'}^*)^\top \boldsymbol{X}) + \beta_{0j'}^* : i' \in [N], j' \in [L] \right\}.$$

Since $g_G(\boldsymbol{X}) = g_{G_*}(\boldsymbol{X})$ for almost every $\boldsymbol{X}$, we have

$$\sum_{i=1}^{N} \text{softmax}_G(\boldsymbol{X}^\top B_i \boldsymbol{X} + c_i^0) \cdot h(\boldsymbol{X}, \eta_i^0) + \sum_{j=1}^{L'} \text{softmax}_G\left((\beta_{1j})^\top \boldsymbol{X} + \alpha\sigma(\tau(\beta_{1j})^\top \boldsymbol{X}) + \beta_{0j}\right) \cdot h(\boldsymbol{X}, \eta_j)$$

$$= \sum_{i=1}^{N} \text{softmax}_{G_*}(\boldsymbol{X}^\top B_i \boldsymbol{X} + c_i^0) \cdot h(\boldsymbol{X}, \eta_i^0) + \sum_{j=1}^{L} \text{softmax}_{G_*}\left((\beta_{1j}^*)^\top \boldsymbol{X} + \alpha\sigma(\tau(\beta_{1j}^*)^\top \boldsymbol{X}) + \beta_{0j}^*\right) \cdot h(\boldsymbol{X}, \eta_j^*). \tag{56}$$

As the expert function $h$ satisfies the conditions in Definition 4.2, the set $\{h(\boldsymbol{X}, \eta_i') : i \in [k']\}$, where $\eta_1', \ldots, \eta_{k'}'$ are distinct vectors for some $k' \in \mathbb{N}$, is linearly independent. If $L' \neq L$, then there exists some $i \in [L']$ such that $\eta_i \neq \eta_j^*$ for any $j \in [L]$. This implies that $\sum_{j=1}^{L'} \text{softmax}_G\left((\beta_{1j})^\top \boldsymbol{X} + \alpha\sigma(\tau(\beta_{1j})^\top \boldsymbol{X}) + \beta_{0j}\right) \cdot h(\boldsymbol{X}, \eta_j) = 0$, which is a contradiction. Thus, we must have that $L = L'$. As a result,

$$\left\{ \text{softmax}_G\left((\beta_{1j})^\top \boldsymbol{X} + \alpha\sigma(\tau(\beta_{1j})^\top \boldsymbol{X}) + \beta_{0j}\right) : j \in [L'] \right\}$$

$$= \left\{ \text{softmax}_{G_*}\left((\beta_{1j}^*)^\top \boldsymbol{X} + \alpha\sigma(\tau(\beta_{1j}^*)^\top \boldsymbol{X}) + \beta_{0j}^*\right) : j \in [L] \right\},$$

for almost every $\boldsymbol{X}$. WLOG, we may assume that

$$\text{softmax}_G\left((\beta_{1j})^\top \boldsymbol{X} + \alpha\sigma(\tau(\beta_{1j})^\top \boldsymbol{X}) + \beta_{0j}\right) = \text{softmax}_{G_*}\left((\beta_{1j}^*)^\top \boldsymbol{X} + \alpha\sigma(\tau(\beta_{1j}^*)^\top \boldsymbol{X}) + \beta_{0j}^*\right), \tag{57}$$

for almost every $\boldsymbol{X}$ for any $j \in [L]$. Since the softmax function is invariant to translation, this result indicates that $\beta_{1j} = \beta_{1j}^*$ and $\beta_{0j} = \beta_{0j}^* + v_0$ for some $v_0 \in \mathbb{R}$ for any $j \in [L]$. Recall from the universal assumption that $\beta_{0L'} = \beta_{0L} = 0$, we get that $\beta_{0j} = \beta_{0j}^*$ for any $j \in [L]$. Then, equation (56) can be rewritten as

$$\sum_{j=1}^{L} \exp(\beta_{0j}) \exp\left((\beta_{1j})^\top \boldsymbol{X} + \alpha\sigma(\tau(\beta_{1j})^\top \boldsymbol{X})\right) h(\boldsymbol{X}, \eta_j)$$

$$= \sum_{j=1}^{L} \exp(\beta_{0j}^*) \exp\left((\beta_{1j}^*)^\top \boldsymbol{X} + \alpha\sigma(\tau(\beta_{1j}^*)^\top \boldsymbol{X}\right) h(\boldsymbol{X}, \eta_j^*), \tag{58}$$

for almost every $\boldsymbol{X}$. Next, we denote $P_1, P_2, \ldots, P_m$ as a partition of the index set $[L]$, where $m \leq L'$, such that $\exp(\beta_{0i}) = \exp(\beta_{0i'}^*)$ for any $i, i' \in P_j$ and $j \in [L]$. On the other hand, when $i$ and $i'$ do not belong to the same set $P_j$, we let $\exp(\beta_{0i}) \neq \exp(\beta_{0i'})$. Thus, we can reformulate equation (58) as

$$\sum_{j=1}^{m} \sum_{i \in P_j} \exp(\beta_{0i}) \exp\left((\beta_{1i})^\top \boldsymbol{X} + \alpha\sigma(\tau(\beta_{1i})^\top \boldsymbol{X}\right) h(\boldsymbol{X}, \eta_i)$$

$$= \sum_{j=1}^{m} \sum_{i \in P_j} \exp(\beta_{0i}^*) \exp\left((\beta_{1i}^*)^\top \boldsymbol{X} + \alpha\sigma(\tau(\beta_{1i}^*)^\top \boldsymbol{X}\right) h(\boldsymbol{X}, \eta_i^*),$$

for almost every $\boldsymbol{X}$. Recall that $\beta_{1i} = \beta_{1i}^*$ and $\beta_{0i} = \beta_{0i}^*$ for any $i \in [L]$, then the above equation leads to

$$\{\eta_i : i \in P_j\} \equiv \{\eta_i^* : i \in P_j\},$$

for almost every $\boldsymbol{X}$ for any $j \in [m]$. As a consequence,

$$G = \sum_{j=1}^m \sum_{i \in P_j} \exp(\beta_{0i})\delta_{(\beta_{1i},\eta_i)} = \sum_{j=1}^m \sum_{i \in P_j} \exp(\beta_{0i}^*)\delta_{(\beta_{1i}^*,\eta_i^*)} = G_*.$$

Hence, we reach the conclusion of this proposition.

## C    Discussion of related Mixture of Experts works

Recently, the MoE model has been employed to mitigate catastrophic forgetting in continual learning. For example, [58] focused on continual learning in vision-language models by adapting a pre-trained vision-language model to new tasks through learning a mixture of specialized adapter modules. [58] introduced an MoE structure onto a frozen CLIP, utilizing a mixture of adapters to modify the MLP block after the MSA layer. In contrast, our work centers on general continual learning with pre-trained models, leveraging the inherent MoE architecture of MSA layers. Consequently, our MoE model placement differs from that of [58]. By employing prefix tuning, we demonstrate that it is analogous to introducing new prefix experts to scale and adapt these pre-trained MoE models to downstream tasks. Furthermore, while [58] utilizes task-specific routers, our approach employs task-specific prompts that encapsulate both task-specific router and expert parameters.

The parameters cost is usually considered in practical memory-constrained continual learning scenarios. Dynamic routing mechanism can be employed for gating-based neural networks [18]. To improve the parameter efficiency of the final model, we can integrate this mechanism in the proposed method. Specifically, each head in the MSA layers comprises $N$ MoE models, where $N$ is the length of the input sequence. This allows for a dynamic routing mechanism to enhance parameter efficiency. For instance, [18] proposed a dynamic routing strategy that adaptively adjusts the number of activated experts based on the input. The computation for any MoE model's gating is directly correlated with the corresponding row in the attention matrix, which encapsulates the MoE model's score functions. For example, selecting the top $k$ experts via Top-K routing in the $i$-th MoE model is equivalent to identifying the top $k$ largest values in the $i$-th row of the attention matrix. To implement [18], we first sort the elements in the $i$-th row from highest to lowest, then find the smallest set of experts whose cumulative probability exceeds the threshold. Consequently, unselected experts remain inactive, reducing the need to compute all elements of the value matrix within self-attention.

## D    Training Algorithm of HiDe-Prompt

In this appendix, we outline the detailed algorithm of HiDe-Prompt, utilizing the same notation as in Section 2.

Each previously encountered class $c \in \mathcal{Y}^{(i)}, i = 1, \ldots, t-1$ has its instructed and uninstructed representations approximated by Gaussian distributions, denoted as $\mathcal{G}_c$ and $\hat{\mathcal{G}}_c$, respectively.

HiDe-Prompt maintains an expandable pool of task-specific prompts $\boldsymbol{e}_t$, each optimized for a specific task $\mathcal{D}_t$ using a cross-entropy loss within the WTP objective. To prevent forgetting, previous prompts $\boldsymbol{e}_1, \ldots, \boldsymbol{e}_{t-1}$ remain frozen. Knowledge transfer across tasks is facilitated by a prompt ensemble (PE) strategy: the current prompt is initialized with the last one $\boldsymbol{e}_t \leftarrow \boldsymbol{e}_{t-1}$ and refined using a weighted combination of all past prompts $\boldsymbol{p}_t = \alpha \sum_{i=1}^{t-1} \boldsymbol{e}_i + (1-\alpha)\boldsymbol{e}_t$, where $\alpha$ is a hyper-parameter. Notably, HiDe-Prompt incorporates contrastive regularization within the WTP objective, pushing features of the new task away from those of past tasks represented by the prototypes of old class distributions $\mathcal{G}_c$. Let $\mathcal{H}_t = \{f_\theta(\boldsymbol{x}_i^{(t)}, \boldsymbol{p}_t) | i = 1, \ldots, N_t\}$ be the embedding transformation of $\mathcal{D}_t$ and $\boldsymbol{\mu}_c$ be the mean of $\mathcal{G}_c$. The contrastive loss can be written as

$$\mathcal{L}_{\text{CR}}(\boldsymbol{p}_t) = \sum_{h \in \mathcal{H}_t} \sum_{i=1}^{t-1} \sum_{c \in \mathcal{Y}^{(i)}} \log\left(\frac{\exp(\boldsymbol{h} \cdot \boldsymbol{\mu}_c / \tau)}{\sum_{\boldsymbol{h}' \in \mathcal{H}_t} \exp(\boldsymbol{h} \cdot \boldsymbol{h}' / \tau) + \sum_{i=1}^{t-1} \sum_{c' \in \mathcal{Y}^{(i)}} \exp(\boldsymbol{h} \cdot \boldsymbol{\mu}_{c'} / \tau)}\right), \quad (59)$$

where $\tau$ is the temperature that is set to 0.8. The overall objective function of WTP for learning a new task $t$ is defined as

$$\mathcal{L}_{\text{WTP}}(\psi, \boldsymbol{p}_t) = \mathcal{L}_{\text{CE}}(\psi, \boldsymbol{p}_t) + \lambda \mathcal{L}_{\text{CR}}(\boldsymbol{p}_t), \quad (60)$$

**Algorithm 1** HiDe-Prompt's training algorithm

---

**Input:** Pre-trained transformer backbone $f_\theta$, training sets $\mathcal{D}_1, \ldots, \mathcal{D}_T$, number of tasks $T$, number of epochs $E$, hyper-parameters $\alpha$, $\tau$ and $\lambda$.
**Output:** Parameters $\boldsymbol{p}_1, \ldots, \boldsymbol{p}_T, \omega$ and $\psi$

1: Initialize $\boldsymbol{e}_1, \omega$ and $\psi$
2: **for** $t = 1, \ldots, T$ **do**
3:     **for** $c \in \mathcal{Y}^{(t)}$ **do**
4:         Obtain $\hat{\mathcal{G}}_c$ from $f_\theta$ and $\mathcal{D}_t$                             $\triangleright$ Uninstructed Representations
5:     **if** $t > 1$ **then**
6:         Initialize $\boldsymbol{e}_t \leftarrow \boldsymbol{e}_{t-1}$
7:         Construct $\boldsymbol{p}_t = \alpha \sum_{i=1}^{t-1} \boldsymbol{e}_i + (1 - \alpha)\boldsymbol{e}_t$
8:     **else**
9:         Construct $\boldsymbol{p}_t = \boldsymbol{e}_t$
10:     **for** $epoch = 1, \ldots, E$ **do**
11:         Optimize $\boldsymbol{p}_t$ and $\psi$ with $\mathcal{L}_{\text{WTP}}$ in Eq.(60)         $\triangleright$ Within-Task Prediction
12:         Optimize $\omega$ with $\mathcal{L}_{\text{TII}}$ in Eq.(62)              $\triangleright$ Task-Identity Inference
13:         Optimize $\psi$ with $\mathcal{L}_{\text{TAP}}$ in Eq.(61)           $\triangleright$ Task-Adaptive Prediction
14:     **for** $c \in \mathcal{Y}^{(t)}$ **do**
15:         Obtain $\mathcal{G}_c$ from $f_\theta, \boldsymbol{p}_t$ and $\mathcal{D}_t$               $\triangleright$ Instructed Representations
      **return** $(\boldsymbol{p}_1, \ldots, \boldsymbol{p}_T, \omega, \psi)$

---

where $\lambda$ is a hyper-parameter. Following WTP, HiDe-Prompt performs a further refinement step on the output layer parameters $\psi$ using a separate objective called task-adaptive prediction (TAP). TAP addresses potential classifier bias by considering the Gaussian distribution of all classes encountered so far. The final output layer $h_\psi$ can be further optimized for TAP objective,

$$\mathcal{L}_{\text{TAP}}(\psi) = \sum_{i=1}^{t} \sum_{c \in \mathcal{Y}^{(i)}} \sum_{\boldsymbol{h} \in \mathcal{H}_{i,c}} -\log\left(\frac{\exp(h_\psi(\boldsymbol{h})[c])}{\sum_{j=1}^{t} \sum_{c' \in \mathcal{Y}^{(j)}} \exp(h_\psi(\boldsymbol{h})[c'])}\right) \tag{61}$$

where $\mathcal{H}_{i,c}$ is constructed by sampling an equal number of pseudo representations from $\mathcal{G}_c$ for $c \in \mathcal{Y}^{(i)}$ and $i = 1, \ldots, t$.

For TII, HiDe-Prompt leverages a lightweight auxiliary output layer $\hat{h}_\omega : \mathbb{R}^D \to \mathbb{R}^T$, to map uninstructed representations directly to task identity. This mapping is learned explicitly through a cross-entropy loss function,

$$\mathcal{L}_{\text{TII}}(\omega) = \sum_{c \in \mathcal{Y}_t} \sum_{\hat{\boldsymbol{h}} \in \hat{\mathcal{H}}_c} -\log\left(\frac{\exp(\hat{h}_\omega(\hat{\boldsymbol{h}})[c])}{\sum_{c' \in \mathcal{Y}_t} \exp(\hat{h}_\omega(\hat{\boldsymbol{h}})[c'])}\right) \tag{62}$$

where $\hat{\mathcal{H}}_c$ is constructed by sampling an equal number of pseudo representations from $\hat{\mathcal{G}}_c$ for $c \in \mathcal{Y}^{(i)}$ and $i = 1, \ldots, t$. Please refer to Algorithm 1 for more details.

## E  Experimental Details

**Datasets.** We use commonly-used datasets in the field of continual learning, including **(1) Split CIFAR-100** [23]: This dataset comprises images from 100 classes. These classes are divided randomly into 10 separate incremental tasks, with each task featuring a distinct set of classes. **(2) Split ImageNet-R** [23]: This dataset is composed of images from 200 classes. It includes challenging examples from the original ImageNet [40] dataset and newly gathered images representing diverse styles. These classes are also randomly divided into 10 distinct incremental tasks. **(3) Split CUB-200** [48]: This dataset consists of fine-grained images of 200 different bird species. It is randomly divided into 10 incremental tasks, each comprising a unique class subset. **(4) 5-Datasets** [9]: This composite dataset incorporates **CIFAR-10** [23], **MNIST** [24], **Fashion-MNIST** [55], **SVHN** [31], and **notMNIST** [3]. Each of these datasets is treated as a separate incremental task, permitting for the assessment of the effects of significant variations between tasks.

**Prompt-Based Approaches.** We compare NoRGa against recent prompt-based continual learning approaches: L2P [54], DualPrompt [53], CODA-Prompt [42], S-Prompt [52] and HiDe-Prompt [49].

Table 4: Performance comparison in task-incremental learning setting. Here we present Final Average Accuracy (FA).

| Method | Split CIFAR-100 | | Split CUB-200 | |
| --- | --- | --- | --- | --- |
| | Sup-21K | iBOT-21K | Sup-21K | iBOT-21K |
| HiDe-Prompt | $97.87 \pm 0.31$ | $97.48 \pm 0.33$ | $97.57 \pm 0.08$ | $92.34 \pm 0.34$ |
| NoRGa tanh | $98.55 \pm 0.45$ | $\mathbf{98.26 \pm 0.36}$ | $97.86 \pm 0.14$ | $92.85 \pm 0.33$ |
| NoRGa sigmoid | $\mathbf{98.63 \pm 0.35}$ | $98.15 \pm 0.29$ | $\mathbf{97.89 \pm 0.14}$ | $92.85 \pm 0.22$ |
| NoRGa GELU | $98.41 \pm 0.47$ | $98.17 \pm 0.30$ | $97.76 \pm 0.10$ | $\mathbf{93.00 \pm 0.11}$ |

Table 5: Performance comparison of different PEFT methods using ViT-B/16 with Sup-21K weights. Here we present Final Average Accuracy (FA).

| Method | Split CIFAR-100 | Split CUB-200 |
| --- | --- | --- |
| HiDe-Prompt | 92.61 | 86.56 |
| HiDe-LoRA | 92.71 | 87.37 |
| HiDe-Adapter | 92.73 | 87.10 |
| NoRGa | **94.48** | **90.90** |

To ensure a fair comparison, we replicate these methods using the configurations reported in their respective papers. S-Prompt in the original paper trains a separate prompt and classifier head for each task. At evaluation, it infers domain id as the nearest centroid obtained by applying K-Means on the training data. To adapt S-Prompt to CIL, we use one common classifier head for all tasks. For NoRGa, we adopt the same configuration as HiDe-Prompt, which utilizes Prefix Tuning [26] as its prompt-based methodology. Learnable scalar factors $\alpha$ and $\tau$ are frozen after the first task's training to mitigate catastrophic forgetting. We further optimize NoRGa by selecting the best non-linear activation function $\sigma$ via cross-validation among $\tanh$, $\mathrm{sigmoid}$, and $\mathrm{GELU}$.

**Evaluation Metric.** We employ three common metrics to measure the performance the methods, including final average accuracy (FA), cumulative average accuracy (CA), and average forgetting measure (FM). Let $S_{i,t}$ denote the accuracy on the $i$-th task after learning the $t$-th task, and $A_t$ represent the average accuracy as $A_t = \frac{1}{t} \sum_{i=1}^{t} S_{i,t}$. Upon learning all $T$ tasks, we compute FA $= A_T$, CA $= \frac{1}{T} \sum_{t=1}^{T} A_t$, and FM $= \frac{1}{T-1} \sum_{i=1}^{T-1} \max_{t \in \{1,...,T-1\}} (S_{i,t} - S_{i,T})$. It's worth noting that FA and CA are prioritized over FM, as they inherently encompass both plasticity and forgetting, with FM providing supplementary context [42].

**Implementation Details.** We train and test on one NVIDIA A100 GPU for baselines and our method. We leverage a pre-trained ViT-B/16 model as the backbone. Training employs an Adam optimizer ($\beta_1 = 0.9$, $\beta_2 = 0.999$), a batch size of 128, and a constant learning rate of 0.005 for all methods except CODA-Prompt. CODA-Prompt utilizes a cosine decaying learning rate starting at 0.001. Additionally, a grid search technique was implemented to determine the most appropriate number of epochs for effective training.

## F  Task-incremental Learning Results

Because HiDe-Prompt optimizes prompt parameters specifically for within-task prediction (WTP), NoRGa inherently aligns with this objective, leading to generally better continual learning performance. We demonstrate this improvement through experiments in a task-incremental learning setting, where task labels are available during inference (as in Table 4). While HiDe-Prompt performs well, NoRGa shows consistent improvement across all scenarios. Notably, NoRGa with sigmoid activation achieves the highest final average accuracy in both Split CIFAR-100 and Split CUB-200 with Sup-21K training. Additionally, NoRGa demonstrates its effectiveness even with self-supervised pre-training, further solidifying its advantage over the original prefix tuning model. Overall, NoRGa variants outperform HiDe-Prompt on both datasets and under both training conditions.

## G  Comparison to Different Parameter-Efficient Fine-Tuning Methods

As the advantages of different parameter-efficient fine-tuning (PEFT) methods remain an open question, we briefly describe them through our revealed connection between self-attention and MoE.

Table 6: Performance comparison of pre-trained model-based continual learning methods using ViT-B/16 with Sup-21K weights. Here we present Final Average Accuracy (FA).

| Method | Split CIFAR-100 | Split CUB-200 |
|---|---|---|
| ADAM + VPT-D | 85.04 | 85.28 |
| ADAM + SSF | 85.27 | 85.67 |
| ADAM + Adapter | 87.29 | 85.84 |
| RanPAC | 92.20 | 90.30 |
| NoRGa | **94.48** | **90.90** |

Table 7: Ablation study on the effect of learnable $\alpha$ and $\tau$ with Sup-21K weights. Here we present Final Average Accuracy (FA).

| Method | Split CIFAR-100 | Split CUB-200 |
|---|---|---|
| HiDe-Prompt | 92.61 | 86.56 |
| Learnable $\alpha$, Fixed $\tau$ | 94.38 | 90.45 |
| Fixed $\alpha$, Learnable $\tau$ | 94.42 | 90.48 |
| Fixed $\alpha$, Fixed $\tau$ | 94.29 | 90.32 |
| Learnable $\alpha$, Learnable $\tau$ | **94.48** | **90.90** |

Prefix tuning introduces additional parameters at the input of MSA layers to adapt the pre-trained model representation, contrasting with Adapter [16], which insert adaptive parameters between layers, often replacing MLP blocks. LoRA [17] approximates weight updates with low-rank matrices and adds them to the backbone weights. Our work shows that the MSA layer in a pre-trained model can be seen as a pre-trained MoE architecture. Applying LoRA to the MSA layer refines both the pre-trained experts and their corresponding score functions for downstream tasks. In contrast, prefix tuning expands the pre-trained MoE models by incorporating new experts while preserving the original components, rather than modifying the pre-trained experts like LoRA.

NoRGa emerges as a simple, parameter-efficient fine-tuning method and can be regarded as a distinct implementation of prompts. However, our novel perspective on the interplay between self-attention, prefix tuning, and mixture of experts enables us to theoretically substantiate the effectiveness of NoRGa as shown in Section 4.

For empirical comparison, we integrate the framework of HiDe-Prompt with different PEFT techniques and Sup-21K weights, evaluating performance on Split CIFAR-100 and Split CUB-200. The results are summarized in Table 5. The table shows that NoRGa consistently outperforms the other PEFT methods on both datasets, suggesting its effectiveness. Nevertheless, further investigation with LoRA and Adapter would be necessary to draw more definitive conclusions.

While exploring alternative PEFT methods might offer improvements in WTP performance, these approaches lack theoretical guarantees and could lead to an increased number of parameters. In contrast, our NoRGa method modifies the original score functions of prefix tuning to enhance WTP performance with theoretical rigor. Importantly, NoRGa maintains the same parameter count as HiDe-Prompt, which is crucial in CL due to memory constraints.

## H Comparison with Pre-trained Model-based Methods

Previous works have demonstrated that utilizing pre-trained models (PTM) significantly enhances performance for continual learning, often surpassing the performance of non-PTM-based methods. Moreover, studies have shown that first-task adaptation and simple PEFT-style tuning can achieve competitive performance [20, 37, 60, 29] with prompt-based methods. For instance, [20] demonstrated that appending a nearest class mean (NCM) classifier to a ViT model's feature outputs, can serve as a strong baseline. [37, 60] enhanced this strategy by adapting the pre-trained model to the first task using the three PEFT methods for transformer networks [60] and the FiLM method for CNNs [37]. Additionally, [29] improved NCM by incorporating second-order statistics—covariance and Gram matrices. However, these methods, which fine-tune only the backbone for the initial task, may not always ensure satisfactory separation of new tasks' features. Our work focuses on continually adapting the backbone, utilizing task-specific prompts to consistently capture emerging

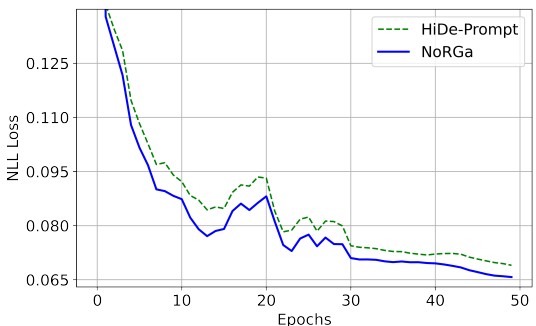

Figure 3: Validation loss on Split CUB-200 throughout the training of the first task.

Table 8: Comparison of training times for HiDe-Prompt and NoRGa. All experiments were conducted on a single NVIDIA A100 GPU.

| Method | Split CIFAR-100 | Split ImageNet-R | Split CUB-200 | 5-Datasets |
|---|---|---|---|---|
| HiDe-Prompt | 2.80h | 2.67h | 1.04h | 24.06h |
| NoRGa | 2.85h | 2.70h | 1.10h | 24.23h |

tasks' characteristics, and proposing a novel method to enhance the CL performance of previous prompting methods.

To validate our approach, we compare it against state-of-the-art PTM-based continual learning methods, including ADAM [60] and RanPAC [29], using Split CIFAR-100 and Split CUB-200 datasets. The results are summarized in Table 6. In comparison to other PTM-based continual learning methods, NoRGa demonstrates competitive performance across both evaluated datasets. For instance, on Split CIFAR-100, NoRGa achieves an FA of 94.48%, exceeding the next best method by over 2%. Similarly, on Split CUB-200, NoRGa delivers strong results relative to other baselines. These improvements highlight the effectiveness of our method in mitigating catastrophic forgetting and preserving knowledge retention across multiple tasks.

## I  Efficiency Tests

We compare the validation loss of NoRGa and HiDe-Prompt throughout the first task on Split CUB-200, as illustrated in Figure 3. The results demonstrate that NoRGa consistently outperforms HiDe-Prompt throughout the training process. This empirical evidence supports the theoretical advantages of NoRGa over HiDe-Prompt.

## J  Effect of Learnable Hyperparameters

As described in Section 4.1 and Appendix E, in our framework, $\alpha$ and $\tau$ are learnable hyperparameters and optimized through backpropagation during the first task, eliminating the need for manual tuning. Additionally, our theoretical analysis of NoRGa's statistical efficiency in Section 4.2 holds for any values of $\alpha$ and $\tau$, demonstrating the theoretical robustness. To further investigate, we evaluated both fixed and learnable settings for these hyperparameters. For the fixed case, we set their values to 1. We report FA on Split CUB-200 and Split CIFAR-100 with Sup-21K weights. The results are summarized in Table 7. Although performance slightly decreased with fixed hyperparameters, it still outperforms HiDe-Prompt, indicating our method's empirical robustness.

## K  Training Times

We utilize a single A100 GPU for all experiments. The training times are summarized in Table 8. While NoRGa exhibits slightly longer training times compared to HiDe-Prompt, it consistently achieves significantly better performance. This demonstrates the effectiveness of NoRGa while maintaining competitive training efficiency.

