# OpenReview forum: "Mixture of Experts Meets Prompt-Based Continual Learning"
_NeurIPS.cc/2024/Conference — NeurIPS 2024 poster_

### Official Review · Reviewer_boEM · 2024-07-09

**Soundness:** 3
**Presentation:** 2
**Contribution:** 3
**Rating:** 6
**Confidence:** 2

**Summary:**

The paper titled "Mixture of Experts Meets Prompt-Based Continual Learning" explores the integration of prompt-based continual learning methods with mixture of experts (MoE) architectures. The paper proposes a novel gating mechanism called Non-linear Residual Gates (NoRGa) to enhance the performance of prompt-based continual learning by leveraging theoretical insights and empirical evidence.

**Strengths:**

1. This paper offers a novel connection between prompt-based-tuning and mixture-of-experts, providing a fresh perspective on prompt-based continual learning approaches.

2. Introduction of NoRGa, which integrates non-linear activation and residual connections to enhance continual learning performance while maintaining parameter efficiency.

**Weaknesses:**

1. Comparing NoRGa with other state-of-the-art continual learning methods that do not use prompts would highlight the specific advantages of the proposed method.

2. It may be better to use a additional graph to represent the final method NoRGa of the paper.

3. It may be necessary to further clarify the differences between the proposed method and other prompt-based methods.

**Questions:**

Which dataset is the ViT-B/16 pre-trained on.

**Limitations:**

This paper has already discussed the limitations of our work in the conclusion section.

---

> ### Author Rebuttal · Authors · 2024-08-03
>
> Thank you for your constructive feedback and insightful comments. Below, we provide a point-to-point response to these comments and summarize the corresponding revisions in final version.
>
> __Q1: Comparing NoRGa with other state-of-the-art continual learning methods that do not use prompts would highlight the specific advantages of the proposed method.__
>
> A1: Thank you for your valuable suggestion. As our work focuses on the continual learning of pretrained models, we have limited our comparison to pretrained model-based (PTM-based) continual learning methods. Previous works have also demonstrated that utilizing pretrained models has shown great promise and performance for continual learning, surpassing the performance upper bound of non-PTM-based methods. Specifically, we compared our method (NoRGa) against ADAM [1], RanPAC [2], and ESN [3], using ViT-B/16 with pretrained Sup-21K weights. Performance was evaluated using final average accuracy (FA) on Split CIFAR-100 and Split CUB-200. The results can be summarized as follows:
>
> | Method         | Split CIFAR-100 | Split CUB-200 |
> |----------------|-----------------|------------------|
> | ADAM + VPT-D [1] | 85.04 | 85.28 |
> | ADAM + SSF [1] | 85.27  | 85.67 |
> | ADAM + Adapter [1] | 87.29  | 85.84 |
> | RanPAC [2] | 92.20 | 90.30 |
> | ESN [3] | 86.34           | N/A |
> | NoRGa (ours) | __94.48__ | __90.90__ |
>
> As shown, NoRGa exhibits competitive performance on both datasets. For example, on Split CIFAR-100, NoRGa achieves an FA of 94.48%, surpassing the next best method by over 2%. On Split CUB-200, NoRGa also demonstrates competitive results compared to other baselines. This improvement underscores the effectiveness of our proposed method in mitigating catastrophic forgetting and preserving knowledge across multiple tasks. A more detailed comparison will be included in the final version.
>
> __Q2: It may be better to use a additional graph to represent the final method NoRGa of the paper.__
>
> A2: Thank you for your valuable suggestion. You can refer to **Q1** in General Response.
>
> __Q3: It may be necessary to further clarify the differences between the proposed method and other prompt-based methods.__
>
> A3: Thank you for your valuable suggestion. We compare the differences between L2P, DualPrompt, HiDe-Prompt, and NoRGa (ours) as follows:
> -  __L2P:__ Utilizes a shared prompt pool for all tasks. Each prompt is associated with a learnable prompt key. We then employ the query feature $q(\boldsymbol{x})$ to retrieve the top K most similar prompts using cosine distance. Consequently, the most relevant keys and corresponding prompts are explicitly assigned to instances based on the query feature.
> -  __DualPrompt:__ Enhances L2P by using two complementary prompts during training: a general prompt (G-Prompt) and a task-specific expert prompt (E-Prompt) per task. The set of E-Prompts acts as an expanding pool of task-specific knowledge, similar to the L2P prompt pool, but with the key difference of growing incrementally with each new task. DualPrompt employs the same prompt selection mechanism as L2P for the E-Prompts. In contrast, the G-Prompt is shared among all tasks, requiring no prompt selection.
> -  __HiDe-Prompt:__ A recent SOTA prompt-based method that employs only task-specific E-Prompts. Prompts for each task are trained with the task's objective and a contrastive regularization that tries to push features of new tasks away from prototypes of old ones. Unlike L2P and DualPrompt, prompt selection is achieved by an additional MLP head placed atop the pre-trained ViT, which employs $q(\boldsymbol{x})$ to determine the suitable prompt.
> -  __NoRGa:__ Utilizes the same framework as HiDe-Prompt. Specifically, NoRGa only uses task-specific prompts (E-Prompts) with an additional MLP head for prompt selection. Recognizing that MSA layers embody a mixture of experts (MoE) architecture and applying prefix tuning is the process of introducing new experts into these models, NoRGa modifies the gating mechanism of prefix tuning, addresses statistical limitations of prefix tuning, and enhances continual learning performance.
>
> We will add the above discussion in the final version.
>
> __Q4: Which dataset is the ViT-B/16 pre-trained on?__
>
> A4: Thanks for your question. The ViT-B/16 model is pre-trained on the ImageNet dataset [4] (ImageNet-1K and ImageNet-21K). We utilize several publicly available checkpoints to demonstrate the effectiveness and robustness of our proposed method under varying pretraining settings:
> - Sup-21K: Supervised pretraining on ImageNet-21K
> - iBOT-21K: Self-supervised pretraining on ImageNet-21K
> - iBOT-1K, DINO-1K, MoCo-1K: Self-supervised pretraining on ImageNet-1K.
>
>
> [1] Revisiting Class-Incremental Learning with Pre-Trained Models: Generalizability and Adaptivity are All You Need, arxiv 2023.
>
> [2] Ranpac: Random projections and pre-trained models for continual learning, NeurIPS 2023.
>
> [3] Isolation and Impartial Aggregation: A Paradigm of Incremental Learning without Interference, AAAI 2023.
>
> [4] ImageNet: A large-scale hierarchical image database, CVPR 2009

---

> > ### Comment · Reviewer_boEM · 2024-08-12
> >
> > I have read the authors' response and my concerns have been addressed. I raise my rating to 6.

---

> > > ### Author Response · Authors · 2024-08-12
> > > **Thank you**
> > >
> > > We would like to thank the reviewer for rating a positive score of 6. We are happy to discuss more if the reviewer still has questions.
> > >
> > > Best regards,
> > >
> > > Authors

---

### Official Review · Reviewer_MKaE · 2024-07-11

**Soundness:** 3
**Presentation:** 3
**Contribution:** 3
**Rating:** 6
**Confidence:** 4

**Summary:**

The paper explores the theoretical underpinnings and practical implications of prompt-based methods in continual learning, aiming to enhance our understanding and optimize their effectiveness. It introduces a novel perspective by connecting prefix tuning with mixture of experts models, revealing insights into how self-attention mechanisms encode specialized architectures. The proposed Non-linear Residual Gates (NoRGa) further advances this by improving within-task prediction accuracy and overall continual learning performance while maintaining parameter efficiency.

**Strengths:**

1. This paper introduces a novel theoretical framework connecting self-attention mechanisms to mixture of experts models, significantly advancing the understanding of prompt-based approaches in continual learning.
2. The theoretical insights are well-supported and complemented by empirical experiments across diverse benchmarks, demonstrating robustness and reliability.
3. The concepts, although complex, are explained with clarity, aided by concrete examples and theoretical justifications.
4.  Addressing the gap in theoretical understanding, the paper proposes a practical enhancement (NoRGa) that promises substantial improvements in model adaptation and efficiency.

**Weaknesses:**

1. The paper could benefit from clearer explanations regarding the core elements of prompt-based methods, particularly in how prompts are defined and utilized within their framework.
2. While NoRGa is presented as a novel gating mechanism, its explicit connection to prompts could be elaborated further. Is it reasonable to interpret NoRGa's gating mechanism as a variant or extension of prompts in a broader sense?

**Questions:**

1. Could NoRGa's gating mechanism be seen as a different interpretation or implementation of prompts, albeit operating at a different level or with distinct functional goals?
2. Are there instances of redundant theoretical explanations or definitions throughout the paper? Streamlining these could improve clarity and focus on the novel contributions without detracting from the foundational concepts.
3. Given the complexity of the concepts, would integrating more visual aids (e.g., diagrams illustrating the architecture of NoRGa or the relationship between prompts and gating mechanisms) enhance the paper's accessibility and readability?

**Limitations:**

The paper's reliance on dense theoretical explanations without adequate visual aids may limit its accessibility. Visual representations could significantly enhance comprehension of complex concepts like the relationship between prompts and NoRGa, potentially broadening the paper's audience and impact.

---

> ### Author Rebuttal · Authors · 2024-08-03
>
> Thank you for your constructive feedback and insightful comments. Below, we provide a point-to-point response to these comments and summarize the corresponding revisions in final version.
>
> __Q1: The paper could benefit from clearer explanations regarding the core elements of prompt-based methods, particularly in how prompts are defined and utilized within their framework.__
>
> A1: Thank you for your valuable suggestion. As detailed in Section 2, within our framework, a prompt is a set of learnable parameters denoted by $\mathbf{P} = [ \mathbf{P}^K, \mathbf{P}^V ] \in \mathbb{R}^{L_p \times d}$. These parameters are utilized to fine-tune the multi-head self-attention (MSA) layer of the pretrained model, where $L_p$ is the prompt length and $d$ is the embedding dimension. Notably, the MSA layer incorporates a specialized architecture comprising multiple mixture of experts (MoE) models. Our approach leverages prompt parameters to refine these models by introducing new prefix experts. Specifically, $\mathbf{P}^V \in \mathbb{R}^{\frac{L_p}{2} \times d}$ encodes parameters for new prefix experts appended to the pretrained MoE models. Correspondingly, $\mathbf{P}^K\in \mathbb{R}^{\frac{L_p}{2} \times d}$ encodes parameters for the associated score functions of these new experts within the MoE models in the MSA layer. We adopt task-specific prompts for each task, ensuring that every task has its own set of experts and score functions. During inference, the task identity is inferred to select the appropriate experts and score functions. We will add the above discussion in the final version.
>
> __Q2: Could NoRGa's gating mechanism be seen as a different interpretation or implementation of prompts, albeit operating at a different level or with distinct functional goals?__
>
> A2: NoRGa's gating mechanism can be regarded as a distinct implementation of prompts. As demonstrated in Section 3, the MSA layer in pretrained models can be regarded as a specialized architecture comprising multiple MoE models. Prefix tuning finetunes these MoE models by introducing new prefix experts, utilizing prompts to encode the parameters of the new experts' components. The score functions for newly introduced experts via prefix tuning are linear functions of the input, resulting in suboptimal sample efficiency for parameter estimation as detailed in Appendix A. To mitigate this statistical limitation, NoRGa refines the score functions associated with the original prefix tuning's new experts by incorporating non-linear activation and residual connections, __substantially enhancing statistical efficiency (polynomial versus exponential) with theoretical guarantees__. We will add the above discussion in the final version.
>
> __Q3: Are there instances of redundant theoretical explanations or definitions throughout the paper? Streamlining these could improve clarity and focus on the novel contributions without detracting from the foundational concepts__
>
> A3: Thanks for your question. Upon careful examination, we found minimal redundancy in the theoretical explanations and definitions presented throughout the paper. Section 2 introduces the foundational concepts of MSA, prefix tuning, and mixture of experts (MoE). Building upon this foundation, Section 3 elucidates the connections among self-attention, prefix tuning, and MoE. Based on these insights, we propose a novel method termed NoRGa to enhance statistical efficiency. Our theoretical analysis considers a regression framework to demonstrate that the non-linear residual gating is more sample efficient than the linear gating in terms of estimating experts. The algebraic independence condition is employed to characterize compatible experts for the non-linear residual gating. Finally, we design a loss function based on Voronoi cells for the convergence analysis of expert estimation in MoE models. Our results indicate that under non-linear residual gating, MoE experts exhibit polynomial-order estimation rates, outperforming the $1/\log^{\tau}(n)$ rate observed in the original prefix tuning design (as detailed in Appendix A).
>
>
> __Q4: Given the complexity of the concepts, would integrating more visual aids (e.g., diagrams illustrating the architecture of NoRGa or the relationship between prompts and gating mechanisms) enhance the paper's accessibility and readability?__
>
> A4: Thank you for your valuable suggestion. You can refer to **Q1** in General Response.

---

> > ### Comment · Reviewer_MKaE · 2024-08-10
> > **Official Comment by Reviewer MKaE**
> >
> > Thank you for the author's positive response.
> > Most of my doubts have been resolved, but in fact I have the same doubts as Reviewer dPHR about "the placement of this work as a Continual Learning contribution". I hope the author will seriously consider whether the proposed method can truly solve the core problems such as catastrophic forgetting from the perspective of continuous learning. In addition, the experimental settings are also compared with some popular PTM-based continuous learning methods.
> > Nevertheless, I also appreciate that this work rethinks the relationship between MoE and prompt from a novel and meaningful perspective, so I still maintain the original score.

---

> > > ### Author Response · Authors · 2024-08-10
> > >
> > > We thank the reviewer for maintaining the positive score “6” of the paper. Regarding the question of whether our proposed method effectively addresses core challenges in CL, particularly catastrophic forgetting, we wish to emphasize that our paper is built on the theory that improved WTP is both a necessary and sufficient condition for enhanced CL performance. This theory, which is supported by prior CL research [1] and the original HiDe-Prompt paper, is fundamental to connecting our contribution to continual learning from the perspective of continous learning.
> > >
> > > While exploring alternative Parameter-Efficient Fine-Tuning (PEFT) methods or incorporating more complex expert models might offer improvements in WTP, these approaches lack theoretical guarantees and could lead to an increased number of parameters. In contrast, our NoRGa method modifies the original score functions of prefix tuning to enhance WTP performance with theoretical rigor. Importantly, NoRGa maintains the same parameter count as HiDe-Prompt, which is crucial in CL due to memory constraints.
> > >
> > > We will incorporate the above discussion in the final version. If you have any further questions, please let us know.
> > >
> > > [1] A theoretical study on solving continual learning, NeurIPS 2022.

---

### Official Review · Reviewer_dPHR · 2024-07-12

**Soundness:** 3
**Presentation:** 3
**Contribution:** 3
**Rating:** 5
**Confidence:** 5

**Summary:**

This paper introduces an extension to prefix tuning, by introducing non-linear residual gating - a simple extension over existing prefix tuning. This non-linear residual gating is supported with theoretical efficiency guarantees (polynomial versus exponential) to better estimate the optimal parameters. When applied to the problem of rehearsal-free continual learning with pretrained models (over four standard benchmarks) on top of the HiDe-Prompt method, the suggested gating provides consistent and partly significant improvements.

**Strengths:**

* To the best of my knowledge in both the Continual Learning and the parameter-efficient finetuning domain, the non-linear residual gating prefix MoE is a novel extension over standard prefix tuning.

* This paper is generally well written, and except for the proofs, quite easy to follow.

* Assuming the regression estimator to be a fair assumption, the statistical sample efficiency improvements of the proposed NoRGa setup versus normal gating is significant (polynomial versus exponential).

* The explicit improvements over HiDE-Prompt are consistent, and often significant.

**Weaknesses:**

My biggest issue with this work is its placement as a Continual Learning contribution, and evaluating it as such.

__[1]__ Firstly, the proposed approach, whilel helping with WTP, is simply an extension on top of prefix tuning - and none of the contributions connect to the continual distribution shift nature. The improved performance is much more likely tied to an improvement in the underlying PEFT approach; which in itself is simply sufficient on standard benchmarks (see Zhou et al. 2023). This is further supported by the fact that the authors freeze both alpha and tau after the first task (see supplementary D). Moreover, on such small benchmarks, introducing two additional tunable hyperparameters does incorporate additional degrees of freedom to overfit to these benchmarks, and makes direct comparison to existing methods difficult.

Consequently, the question needs to be answered: How robust is NoRGa with respect to the additional introduced alpha and temperature? These are strictly two hyperparameters more than HiDe-Prompt. As such, how much more compute went into hyperparameter optimization for NoRGa w.r.t. Hide? And given the same search budget, how does HiDe-Prompt compare relatively? This is the most crucial aspect, as the other method comparisons are somewhat redundant, since HiDe-Prompt introduces orthogonal classifier head alignment which can be applied to all other methods as well.

> It would be great if the authors could address both the placement as a continual learning contribution (as opposed to a parameter-efficient finetuning paper, which would need to be evaluated on other respective benchmark tasks), and the comparability to HiDe-Prompt.

__[2]__ This paper also only tackles prompt-based rehearsal-free continual learning, but misses discussions of recent works on first-task-adaptation and simple PEFT-style tuning for Continual Learning, such as: [1] Jason et al. 2022, “A simple baseline that questions the use of pretrained models for continual learning”, [2] Panos et al 2023, “First session adaptation: A strong replay-free baseline for class-incremental learning”, [3] Zhou et al. 2023, “Revisiting class-incremental learning with pretrained models: Generalizability and adaptivity are all you need”, [4] McDonnell et al. 2023: “RanPAC: Random projections and pretrained models for continual learning”.

> Given that these works show that existing prompt-style CL objectives are matched by simply training on the first task, and how performance is matchable with simple parameter-efficient finetuning without explicitly accounting for the continual nature of the problem, how should the insights in this paper be rated w.r.t. These works?

__[3]__ A vast part of this paper deals with the sample in-efficiency of standard prefix tuning mechanisms. However, the authors do not provide any efficiency tests (e.g. how does performance vary as a function of training iterations / samples seen), and only provide slightly improved performances on a few smaller-scale benchmarks. This is somewhat unfortunate, as it would be great to see how the efficiency estimates (e.g. Eq. 16) connect to practical convergence behaviour (e.g. L288 “polynomial number of data versus exponential number of data”).

__[4]__ Both statistical efficiency proofs also do not take into account the continual nature of the problem, but rather assume a single generative model and i.i.d. Samples from said model, which are presented to the learner. This fully disconnects from the actual continual nature and sequential distribution shifts expected in such problems. Similarly, I’m having general troubles to connect the assumptions made throughout the proofs to the practical nature of the problem. E.g. why is Eq. 13 a reasonable assumption for the CL problem? Is convergence behaviour of a LS regression estimator suitably connected to practical convergence behaviour? If so, why? If not, why not? Similarly, why would I want to encourage algebraic independence - what would this give within the actual continual learning scenario?

---

__Smaller issues:__

* I find it somewhat difficult to follow the proofs in 4.2 and Appendix A. Providing a proof-sketch as an overview would significantly help readability.
* It is a bit problematic to have the entire proof of convergence, without even a proof sketch, in the supplementary-  as it motivates the entire section 4 and the non-linear residual gating. It would be great if the authors could provide at the very least a proof sketch in the main paper.
* L139/140 is fairly hard to parse.

**Questions:**

All relevant questions are incorporated in the previous section.

I am currently leaning towards rejection - not because of the proposed method and the theoretical support (both of which I believe are sensible and meaningful), but rather its placement and consequent evaluation as a continual learning method. Neither in the methodological nature NOR the actually conducted proofs do the author account for the continual nature of the problem.
Instead, I believe that this paper is much better suited as a contribution in the domain of parameter-efficient model finetuning. However, this in turn requires different benchmark evaluations.

Moreover, I am uncertain about the significance of the report results (see above).

Together, I would be happy to raise my score if the authors can address these particular concerns (as well as those listed above).

**Limitations:**

The authors do not discuss limitations or societal impact explicitly.

---

> ### Author Rebuttal · Authors · 2024-08-05
>
> Thank you for your constructive feedback and insightful comments. Below, we provide a point-to-point response and summarize the corresponding revisions in the final version.
>
> __Q1: The placement as a continual learning contribution__
>
> A1: Our contributions encompass the introduction of a novel connection between self-attention, prefix tuning, and MoE. __This offers a fresh perspective to the design of previous prompt-based continual learning methods__. For example, DualPrompt uses two complementary prompts during training: a general prompt (G-Prompt) shared across tasks and a task-specific expert prompt (E-Prompt) per task. As illustrated in our paper, each head in the MSA layer comprises multiple MoE models. We hypothesize that the G-Prompt aims to expand the set of pretrained experts in the MSA layer, which capture generalizable knowledge, with new experts encoded within it. However, unlike pretrained experts, the experts in the G-Prompt are learnable across tasks, potentially leading to catastrophic forgetting. Conversely, E-Prompt encodes task-specific experts capturing each task's knowledge.
>
> Furthermore, we agree that NoRGa can have a broader impact to other domain, such as parameter-efficient model finetuning. However, our work builds on HiDe-Prompt, which uses task-specific prompts, i.e. each task has its own set of experts and score functions. NoRGa modifies the original score functions of prefix tuning, enhance WTP performance with theoretical guarantee. We then use the theory that improved WTP performance is sufficent and necessary condition for improved CL performance. This theory is also discussed in previous continual learning works and the original paper of HiDe-Prompt. This theory is critical and connects our contribution to continual learning. Importantly, NoRGa maintains the same parameter count as HiDe-Prompt, which is crucial in continual learning because of the memory constraint.
>
> __Q2: The introduction of 2 hyperparameters $\alpha$ and $\tau$__
>
> A2: In our framework, $\alpha$ and $\tau$ are learnable hyperparameters optimized through backpropagation by the objective of the first task, eliminating the need for manual tuning. __Moreover, our theory on NoRGa's statistical efficiency holds for any values of $\alpha$ and $\tau$, demonstrating the theoretical robustness__. We also experimented with fixed and learnable settings for $\alpha$ and $\tau$. For fixed hyperparameters, we set their values to 1. We report FA on Split CUB-200 and Split CIFAR-100 with Sup-21K weights. The results are summarized below:
>
> |Method|Split CIFAR-100|Split CUB-200|
> |----------------------------|-----------------|---------------|
> |HiDe-Prompt|92.61|86.56|
> |Learnable $\alpha$, $\tau = 1$|94.38|90.45|
> |$\alpha = 1$, Learnable $\tau$|94.42|90.48|
> |$\alpha = 1$, $\tau = 1$|94.29|90.32|
> |NoRGa|__94.48__|__90.90__|
>
> Although performance slightly decreased with fixed hyperparameters, it still outperforms HiDe-Prompt, indicating our method's empirical robustness. We will add this discussion in the final version.
>
> __Q3: Discussions of recent works on first-task-adaptation and simple PEFT-style tuning for Continual Learning__
>
> A3: Thank you for highlighting these excellent related works. We will add the following discussion to the final version:
>
> Previous works have shown that first-task adaptation and simple PEFT-style tuning can achieve competitive performance [1,2,3,4] with prompt-based methods. For instance, [1] demonstrated that appending a nearest class mean (NCM) classifier to a ViT model's feature outputs, can serve as a strong baseline. [2, 3] enhanced this strategy by adapting the pretrained model to the first task using the three PEFT methods for transformer networks [3] and the FiLM method for CNNs [2]. Additionally, [4] improved NCM by incorporating second-order statistics—covariance and Gram matrices. However, these methods, which finetune only the backbone for the initial task, may not always ensure satisfactory separation of new tasks' features. Our work focuses on continually adapting the backbone, utilizing task-specific prompts to consistently capture emerging tasks' characteristics, and proposing a novel method to enhance the CL performance of previous prompting methods.
>
> __Q4: Efficiency tests__
>
> A4: Thank you for your suggestion. We have added the graph to compare Validation loss of NoRGa and HiDe-Prompt throughout the first task in the attached pdf of General Response.
>
> __Q5: Statistical efficiency proofs__
>
> A5: Thanks for your questions. We use the HiDe-Prompt framework with task-specific prompts, where each task has its own experts and score functions optimized with only data from that task. Thus, it is reasonable to assume that samples within a task are generated from a single model and are i.i.d.
>
> In Section 3.2, we show that the output attention head with prefix tuning can be expressed as a linear gating prefix MoE model. Recent works [a, b] have shown that the performance of MoE models can be improved by using a more sample efficient gating function in terms of parameter and expert estimation. Motivated by this, we propose using the non-linear residual gating prefix MoE in Section 4 and conduct the convergence analysis to justify its sample efficiency.
>
> Next, Eq.(13) is a common regression framework to study the sample efficiency of the gating function in MoE models [c], rather than an assumption. Finally, the algebraic independence condition characterizes which experts are compatible with the non-linear residual gating. It indicates that under the non-linear residual gating MoE, experts would have estimation rates of polynomial orders rather than of order $1/\log^{\tau}(n)$ when using the linear gating in Appendix A
>
> [a] Is Temperature Sample Efficient for Softmax Gaussian Mixture of Experts?, ICML 2024
>
> [b] A General Theory for Softmax Gating Multinomial Logistic Mixture of Experts, ICML 2024
>
> [c] On Least Square Estimation in Softmax Gating Mixture of Experts, ICML 2024

---

> > ### Comment · Reviewer_dPHR · 2024-08-09
> > **Re: Rebuttal**
> >
> > I thank the authors for their detailed feedback, which has helped clarified some confusion I had - particularly regarding the placement of the proposed approach in the continual learning literature. Some questions however still remain:
> >
> > * If I understand the reply correctly, this paper directly builds on top of HiDE-Prompt; but does not modify any of the CL components (referring to elements that explicitly account for the continual nature; i.e. the task-specific prompts); but with an explicit focus on improving the WTP (within-task performance). But how is what the authors propose then not just simple parameter-efficient finetuning - simply applied to simplistic CL benchmarks on top of HiDE-Prompt?
> >
> > * Consequently, following Zhou et al. 2023 [3]: What would happen if other PEFT approaches are used to optimize the WTP performance? Or more generally, what does the comparison to first task adaptation sanity checks look like (which are void of meaningful CL elements)? Or a comparison directly to e.g. Zhou et al? Just stating that _"However, these methods, which finetune only the backbone for the initial task, may not always ensure satisfactory separation of new tasks' features."_ may not be sufficient.

---

> > > ### Author Response · Authors · 2024-08-09
> > >
> > > Thank you for your valuable and constructive comments. Here we explain your additional questions as below:
> > >
> > > We agree that NoRGa can serve as a simple, parameter-efficient fine-tuning technique. However, our contributions extend beyond this by offering a novel perspective on the interplay between self-attention, prefix tuning, and the mixture of experts. This offers a fresh viewpoint on the design of previous prompt-based continual learning methods. Furthermore, this relationship enables us to theoretically substantiate the effectiveness of NoRGa.
> > >
> > > While we acknowledge that other PEFT methods, such as LoRA and adapters, could be explored to optimize WTP performance, the relative advantages of different PEFT approaches remain an open question. In contrast, NoRGa not only demonstrates empirical superiority over prefix tuning but also offers theoretical guarantees of enhanced performance. Notably, our theoretical framework imposes no assumptions on the pretrained weights, rendering our method robust across various pretrained models. We have addressed a comparison of different PEFT methods in our response to Reviewer cCSa. Specifically, we employed the HiDe-Prompt framework with various PEFT techniques and Sup-21K weights, evaluating performance using FA on Split CIFAR-100 and Split CUB-200. The results are as follows:
> > >
> > > |Method| Split CIFAR-100| Split CUB-200|
> > > |--------------|-----------------|---------------|
> > > |HiDe-Prompt|92.61|86.56|
> > > |HiDe-LoRA|92.71|87.37|
> > > |HiDe-Adapter|92.73|87.10|
> > > |NoRGa|__94.48__|__90.90__|
> > >
> > > These results highlight the effectiveness of our proposed method relative to other PEFT techniques, although further research is necessary to draw more definitive conclusions. Additionally, we compared our method with a first task-adaptation based method [3], with results summarized below:
> > >
> > > | Method| Split CIFAR-100 | Split CUB-200 |
> > > |----------------|-----------------|------------------|
> > > | ADAM + VPT-D [3]| 85.04 | 85.28 |
> > > | ADAM + SSF [3]| 85.27  | 85.67 |
> > > | ADAM + Adapter [3]| 87.29  | 85.84 |
> > > | NoRGa| __94.48__ | __90.90__ |
> > >
> > > As illustrated, NoRGa achieves the highest performance across both datasets. For instance, on Split CIFAR-100, NoRGa attains an FA of 94.48%, surpassing the next best method by over 7%. This substantial improvement underscores the efficacy of our proposed method in addressing catastrophic forgetting and preserving knowledge across multiple tasks.
> > >
> > > We thank you again for the valuable feedback. If you have any further questions, please let us know.

---

> > > > ### Author Response · Authors · 2024-08-13
> > > > **Any other questions?**
> > > >
> > > > Dear Reviewer dPHR,
> > > >
> > > > We thank you again for your valuable and constructive comments. We already responded to your additional questions. Given that the discussion deadline is approaching (less than 36 hours), can you let us know if our responses sufficiently address your concerns and whether you still have any other questions about the paper?
> > > >
> > > > We thank you again for the valuable feedback.
> > > >
> > > > Best,
> > > >
> > > > The Authors

---

> > > > > ### Comment · Reviewer_dPHR · 2024-08-13
> > > > > **Response to Response**
> > > > >
> > > > > I thank the authors for commenting on my previous response, and providing the additional experimental pointers.
> > > > >
> > > > > The additional experiments comparing different PEFT methods within the directly comparable HiDE framework, and the comparison to first-stage adaptation baselines provide good initial pointers of the strong relative performance of NorGA.
> > > > >
> > > > > While I still believe that NorGA is primarily a parameter-efficient finetuning contribution, the strong performance within an established CL method is quite convincing; though only evaluated on two datasets. I fully understand that the rebuttal timeframe does not provide a lot of opportunities for extended experimental studies, but would love to see a more extensive comparison at least for the final submission.
> > > > >
> > > > > Still, I have consequently decided to raise my score and to recommend acceptance.

---

> > > > > > ### Author Response · Authors · 2024-08-13
> > > > > > **Thank you**
> > > > > >
> > > > > > Thank you for your reply. We sincerely appreciate your endorsement and the time you have taken to provide feedback on our work, which has helped us greatly improve its clarity, among other attributes. Following your suggestion, we will include additional baselines and stronger evaluation comparisons in our revision.
> > > > > >
> > > > > > Best regards,
> > > > > >
> > > > > > Authors

---

### Official Review · Reviewer_cCSa · 2024-07-16

**Soundness:** 2
**Presentation:** 3
**Contribution:** 3
**Rating:** 6
**Confidence:** 5

**Summary:**

The topic of this paper is about the prompt-based continual learning. The authors give a theoretical analysis on these prompt-based continual learning methods, and utilize a Mixture-of-Expert (MoE) architecture characterized by linear experts and quadratic gating score functions. They develop a gating mechanism Non-linear Residual Gates (NoRGa) for MoE-based continual learning. The proposed method has been evaluated on several benchmarks.

**Strengths:**

+ The paper is well-written and easy to follow.
+ It is interesting to theoretically analyze the effectiveness of prompt-based continual learning.
+ Continual learning via MoE architecture is worth exploring.

**Weaknesses:**

-	Comparison to different Parameter-Effiecient-FineTuning methods [b] (e.g. adapter) is needed. This paper mainly focuses on the theoretical analysis of prompt-based continual learning methods.  Prompt-based continual learning belongs to PEFT methods and also add new parameters for new tasks. Can the authors further analyze the advantages of prompt-based methods theroretically?
-	The parameters cost is usually considered in practical memory-constrained continual learning scenarios. Dynamic routing mechanism can be employed for gating-based neural networks (e.g. [c]). To improve the parameter efficiency of the final model, how to integrate this mechanism in the proposed method?
-	Important MoE-related continual learning methods are not included in the related works [a]. The difference between the proposed method and other related works should be highlighted.
-	Performance improvement on some datasets is limited. For example, FA metric (75.06%->75.40%) on Split ImageNet-R, CA metric (95.02%->95.11%) on Split-CIFAR-100 in Table 1. What are the running times of each experiment? Can the authors provide the performance variance of experimental results (e.g. the accuracy at the last incremental learning session)?

[a] Boosting Continual Learning of Vision-Language Models via Mixture-of-Experts Adapters, CVPR2024

[b] Continual Learning with Pre-Trained Models: A Survey, IJCAI2024

[c] Harder Tasks Need More Experts: Dynamic Routing in MoE Models, arxiv2024

**Questions:**

My major concerns are included in the above weaknesses.

**Limitations:**

The authors have addressed the limitations.

---

> ### Author Rebuttal · Authors · 2024-08-03
>
> Thank you for your constructive feedback and insightful comments. Below, we provide a point-to-point response to these comments and summarize the corresponding revisions in final version.
>
> __Q1: Comparison to Different Parameter-Efficient Fine-Tuning Methods and Theoretical Analysis__
>
> A1: Thank you for your valuable suggestion. As the advantages of different PEFT methods remain an open question, we briefly describe them through our revealed connection between self-attention and MoE. Prefix tuning introduces additional parameters at the input of MSA layers to adapt the pretrained model representation, contrasting with adapters, which insert adaptive parameters between layers, often replacing MLP blocks. LoRA approximates weight updates with low-rank matrices and adds them to the backbone weights. Our work shows that the MSA layer in a pretrained model can be seen as a pretrained MoE architecture. Applying LoRA to the MSA layer refines both the pretrained experts and their corresponding score functions for downstream tasks. In contrast, **prefix tuning expands the pretrained MoE models by incorporating new experts while preserving the original components, rather than modifying the pretrained experts** like LoRA.
>
> For empirical comparison, we used the framework of HiDe-Prompt with different PEFT techniques and Sup-21K weights, evaluating performance using FA on Split CIFAR-100 and Split CUB-200. The results are summarized below:
>
> |Method| Split CIFAR-100| Split CUB-200|
> |--------------|-----------------|---------------|
> |HiDe-Prompt|92.61|86.56|
> |HiDe-LoRA|92.71|87.37|
> |HiDe-Adapter|92.73|87.10|
> |NoRGa|__94.48__|__90.90__|
>
> The table shows that NoRGa consistently outperforms the other PEFT methods on both datasets, suggesting its effectiveness. However, further investigation with LoRA and adapters would be necessary to draw more definitive conclusions. We will add this discussion to the final version.
>
> __Q2: Dynamic routing mechanism can be employed for gating-based neural networks (e.g. [c]). To improve the parameter efficiency of the final model, how to integrate this mechanism in the proposed method?__
>
> A2: Thank you for pointing out these excellent related works. We will add the following discussion to the final version:
>
> Each head in the MSA layers comprises $N$ MoE models, where $N$ is the length of the input sequence. This allows for a dynamic routing mechanism to enhance parameter efficiency. For instance, [c] proposed a dynamic routing strategy that adaptively adjusts the number of activated experts based on the input. The computation for any MoE model’s gating is directly correlated with the corresponding row in the attention matrix, which encapsulates the MoE model’s score functions. For example, selecting the top $k$ experts via Top-K routing in the $i$-th MoE model is equivalent to identifying the top $k$ largest values in the $i$-th row of the attention matrix. To implement [c], we first sort the elements in the $i$-th row from highest to lowest, then find the smallest set of experts whose cumulative probability exceeds the threshold. Consequently, unselected experts remain inactive, reducing the need to compute all elements of the value matrix within self-attention.
>
> __Q3: Important MoE-related continual learning methods are not included in the related works [a]. The difference between the proposed method and other related works should be highlighted__
>
> A3: Thank you for providing these excellent related works. We will add the following discussion to the final version:
>
> Recently, the MoE model has been employed to mitigate catastrophic forgetting in continual learning (CL). For example, [a] focused on continual learning in vision-language models by adapting a pretrained vision-language model to new tasks through learning a mixture of specialized adapter modules. [a] introduced an MoE structure onto a frozen CLIP, utilizing a mixture of adapters to modify the MLP block after the MSA layer. In contrast, our work centers on general continual learning with pretrained models, leveraging the inherent MoE architecture of MSA layers. Consequently, our MoE model placement differs from that of [a]. By employing prefix tuning, we demonstrate that it is analogous to introducing new prefix experts to scale and adapt these pretrained MoE models to downstream tasks. Furthermore, while [a] utilizes task-specific routers, our approach employs task-specific prompts that encapsulate both task-specific router and expert parameters.
>
> __Q4: Performance improvement on some datasets is limited.__
>
> A4: While performance gains on certain metrics may be modest for some datasets, our method consistently outperforms the baseline, HiDe-Prompt, the current state-of-the-art in prompt-based continual learning, in terms of either FA or CA. For example, on the Split Imagenet-R dataset with Sup-21K weights, the improvement in FA is small (75.06%->75.40%), but the CA enhancement is significant (76.60%->79.52%). __This trend is consistent across various datasets and pretrained settings, underscoring our method's effectiveness and robustness.__
>
> __Q5: Running Times and Performance Variance__
>
> A5: We utilize a single A100 GPU for all experiments. The training times are summarized below:
>
> | Method| Split CIFAR-100| Split ImageNet-R| Split CUB-200| 5-Datasets|
> |-------------|-----------------|------------------|---------------|------------|
> |HiDe-Prompt|2.80h|2.67h|1.04h|24.06h|
> |NoRGa|2.85h|2.70h|1.10h|24.23h|
>
> Each experiment was conducted 3 times. While NoRGa exhibits slightly longer training times compared to HiDe-Prompt, it consistently achieves significantly better performance as indicated in Table 1. This demonstrates the effectiveness of NoRGa while maintaining competitive training efficiency. We will add the above discussion in the final version. Regarding performance variance, we have already presented the standard deviation of the results in the main results, as displayed in Table 1 of the main text.

---

> > ### Comment · Reviewer_cCSa · 2024-08-12
> > **Rebuttal Response**
> >
> > Thank the authors for their response. Since the authors have addressed all of my concerns, I decided to increase my score.

---

> > > ### Author Response · Authors · 2024-08-13
> > > **Thank you**
> > >
> > > We would like to thank the reviewer for rating a positive score of 6. We are happy to discuss more if the reviewer still has questions.
> > >
> > > Best regards,
> > >
> > > Authors

---

### Author Rebuttal · Authors · 2024-08-04

**General Response:**

We thank all reviewers for their valuable feedback and suggestions, which have significantly contributed to the enhancement of our manuscript. We are encouraged by the endorsements that:
1. Our reveal relationship between self-attention, prefix tuning and mixture of experts is novel, significantly advancing the understanding of prompt-based approaches in continual learning (reviewer MKaE, boEM, cCSa).
2. NoRGa, which integrates non-linear activation and residual connections, can enhance continual learning performance while maintaining parameter efficiency (reviewer boEM)
3. The statistical sample efficiency improvements of NoRGa versus normal gating is significant, promises substantial gains in model adaptation and efficiency (reviewer dPHR, MKaE)
4. The theoretical insights are well-supported and complemented by empirical experiments across diverse benchmarks, demonstrating robustness and reliability (reviewer MKaE). The explicit improvements over HiDE-Prompt are consistent, and often significant (reviewer dPHR).

We address a common comment from Reviewers:

**Q1: Additional graphs/visual aids to represent NoRGa (Reviewer MKaE, boEM)**

**Answer**: We have added some visualizations in the attached PDF. In this file, you can find visual aids illustrating the relationships between self-attention, prefix tuning, and MoE, as well as the NoRGa implementation. We plan to include these in the final manuscript, and we hope they will enhance the paper's accessibility and readability.

We have addressed all the weaknesses and questions raised by all reviewers in the respective rebuttals. We believe that most of this input is valid and greatly improves our paper. We hope that our responses to the reviewer's questions and the additional experiments will help in the review process. Please let us know if you have any further questions.

Regards,

Authors

---

### Decision · Program_Chairs · 2024-09-25

**Decision:**

Accept (poster)

**Comment:**

Prior to rebuttal, the reviewers raised concerns regarding:

1. lack of good connection to the continual nature of CL, specifically, addressing distribution shifts in the proposed approach and the proposed theoretical grounding
2. comparison to PEFT-based CL methods
3. comparison to MoE based CL methods
4. parameter efficiency
5. limited improvement in some of the experiments
6. missing baselines
7. clarity.

the authors seem to have addressed all concerns in their rebuttal, all reviewers either increased or kept the acceptance scores.
In light of the above AC recommends acceptance and urges the authors to incorporate all the reviewer comments and subsequent discussions in the final verios of the paper.